# Real-space spectral simulation of quantum spin models: Application to generalized Kitaev models

Francisco M. O. Brito[⋆] and Aires Ferreira[†]

School of Physics, Engineering and Technology and York Centre for Quantum Technologies,
University of York, York YO10 5DD, United Kingdom

⋆ fmob500@york.ac.uk , † aires.ferreira@york.ac.uk

## Abstract

The proliferation of quantum fluctuations and long-range entanglement presents an outstanding challenge for the numerical simulation of interacting spin systems with exotic ground states. Here, we present a toolset of Chebyshev polynomial-based iterative methods that provides a unified framework to study the thermodynamical properties, critical behavior and dynamics of frustrated quantum spin models with controlled accuracy. Similar to previous applications of the Chebyshev spectral methods to condensed matter systems, the algorithmic complexity scales linearly with the Hilbert space dimension and the Chebyshev truncation order. Using this approach, we study two paradigmatic quantum spin models on the honeycomb lattice: the Kitaev-Heisenberg (K-H) and the Kitaev-Ising (K-I) models. We start by applying the Chebyshev toolset to compute nearest-neighbor spin correlations, specific heat and entropy of the K-H model on a 24-spin cluster. Our results are benchmarked against exact diagonalization and a popular iterative method based on thermal pure quantum states. The transitions between a variety of magnetic phases, namely ferromagnetic, Néel, zigzag and stripy antiferromagnetic and quantum spin liquid phases are obtained accurately and efficiently. We also determine the temperature dependence of the spin correlations, over more than three decades in temperature, by means of a finite temperature Chebyshev polynomial method introduced here. Finally, we report novel dynamical signatures of the quantum phase transitions in the K-I model. Our findings suggest that the efficiency, versatility and low-temperature stability of the Chebyshev framework developed here could pave the way for previously unattainable studies of quantum spin models in two dimensions.

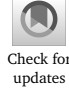

# 1 Introduction

In strongly correlated materials, the interplay between different types of interactions can engender rich $T = 0$ phase diagrams characterized by a series of transitions between paramagnetic, magnetically ordered and quantum spin liquid phases. These quantum phase transitions are driven by one or more parameters in the Hamiltonian, such as an external magnetic field or a spin exchange coupling constant. When a drastic change in the ground state occurs as one of these parameters is varied, there is an accompanying change in the thermodynamic properties. This change manifests itself in the form of critical behavior of quantities such as the structure factor and the susceptibility, which scale with the parameter that drives the transition [1, 2].

The study of the competition between quantum fluctuations and interactions at the heart of quantum phase transitions often calls for a numerical approach [3–11]. Exact solutions are only known for a handful of cases (a well-known example is the isotropic Heisenberg chain [12]). Moreover, in low dimensions, the presence of strong quantum fluctuations limits the applicability of mean-field approaches. Exact diagonalization (ED) methods, such as those based on the Lanczos algorithm [13], are the next step beyond exact analytical solutions. There are several examples of the application of this method to interacting spin systems, for example Refs. [3, 11, 14]. Unfortunately, these are limited to relatively small system sizes, even when algorithms are optimized to reflect symmetries of the model. The culprit is the exponential scal-

ing of the computational cost with the system size, which is particularly severe in dimensions greater than one. Beyond the Lanczos algorithm and its more recent variants [15–17], several attempts to go beyond the limitations of full exact diagonalization have been made. Potent numerical techniques have been deployed with varying degrees of success, including series expansions [18–24], quantum Monte Carlo (QMC) [7,8,25,26], density matrix renormalization group (DMRG) [27–30], tensor-network approaches (such as iPEPS) [10,31–33] and thermal pure quantum (TPQ) states [32,34–36]. Efficient numerical schemes amenable to large-scale computations share a key feature: they aim at reconstructing expectation values of quantum observables without having to fully diagonalize the Hamiltonian. The resulting computational cost depends crucially on how the expectation values of the observables are evaluated. Here, two relevant aspects are at play. The first has to do with how the corresponding operators are reconstructed. The second relates to the process by which one obtains the expectation value. Usually this process is a stochastic one, unless there is prior knowledge about some of the system's features, in which case a variational approach can be viable [25,37,38].

In principle, QMC methods can be used to probe large systems in any number of dimensions, while remaining numerically exact [7,8]. However, they often encounter the so-called sign problem [26,39]. This is a situation where the variance of the estimators of quantities of interest increases exponentially due to quantum statistics. The severity of the problem depends on the computational basis used to tackle the specific model [40–46]. Generally, the sign problem tends to be more acute in frustrated systems [47,48], hampering the use of QMC to extract quantities of interest, such as correlation functions. The sign problem and the limited range of models that QMC is able to access emphasize the need for a general purpose method that can be used more broadly as an alternative to Lanczos ED and QMC.

In this work, we apply a toolset of spectral methods to compute both static (e.g. spin correlations) and dynamic (e.g. spin susceptibility) quantum observables in paradigmatic frustrated systems with competing interactions. Spectral methods based upon Chebyshev expansions have recently proven useful in different contexts [49–60], and here we will be interested in extending this approach to models of interacting quantum spins. We focus on generalized honeycomb Kitaev models, i.e. systems that combine Kitaev interactions with other types of magnetic exchange. Specifically, we study the Kitaev-Heisenberg (K-H) model and the Kitaev-Ising (K-I) model [3–5,61]. The Kitaev model on the honeycomb lattice is one of the rare examples of an exactly solvable microscopic model [62] showing exchange frustration, i.e. nearest neighbor (NN) interactions that cannot be simultaneously minimized. This is similar to geometric frustration which notably occurs in the case of the antiferromagnetic Ising model on the triangular lattice [63]. In the Kitaev case, frustration is created by bond-directional interactions which give way to a fractionalized excitation spectrum of Majorana fermions. This has attracted great attention because it opens up the possibility of synthesizing spin liquid materials with exotic topological orders [64,65]. Notably, in honeycomb iridates, which are transition metal oxides with partially filled d-shells, a subtle interplay of spin–orbit coupling and electronic correlations produces the type of bond-directional interactions that appear in the Kitaev model. Thus, these materials make good spin liquid candidates. In these spin-orbit assisted Mott insulators, the Kitaev exchange interaction is thought to be responsible for the emergence of a spin liquid phase [3,65–68].

The Kitaev exchange interaction also plays a key role in the modelling of other compounds such as the van der Waals ruthenate $\alpha$-RuCl$_3$. A recent study proposes a minimal microscopic 2D spin model for $\alpha$-RuCl$_3$ [69]. The model at play is an extension of the Kitaev model that considers both Kitaev and Heisenberg exchange interactions, third neighbor exchange and $\Gamma$ interactions (i.e., terms that couple different spin components for each nearest neighbor bond). The authors treat this generalized K-H model using a a mean-field random-phase approximation, aiming at extracting quantities such as the dynamical structure factor [69]. The

use of this mean-field approach is justified by comparing its results with those of exact diagonalization [14]. However, exact diagonalization is limited to relatively small system sizes (for example, in Ref. [14], a 24-site cluster is used). Thus, this approach runs into the risk of overlooking large scale properties of the model. In fact, in Refs. [70, 71], the authors detect finite-size effects when computing the dynamical spin structure factor using QMC simulations of the Kitaev model in the presence of disorder — which deems the study of large scale properties crucial — even for clusters of 288 sites.[1] These developments illustrate the need to develop accurate, general purpose computational methods that scale favorably with the system size.

The Chebyshev polynomial methods we shall introduce below have a comparable complexity to Lanczos-based methods. They both share the advantage of being free of the sign problem, even though they can't reach the same system sizes as QMC. Yet, Chebyshev expansions have a few key features that can make them more advantageous than their Lanczos counterparts, namely superior robustness and accuracy. For example, the finite temperature Lanczos method [15] has a low-temperature counterpart [16] that was developed to tackle loss of accuracy due to statistical convergence issues at low-temperature. In this work, we develop a seamless Chebyshev approach that is accurate and does not require a low-temperature counterpart. Moreover, as we will show below, it has advantages even over TPQ [34, 35]. Another application is to study dynamics (e.g., spectral functions), where it proves more flexible and efficient than its Lanczos-based counterpart [17].

This paper is organized as follows. Section 2 provides a bird's-eye view of the Chebyshev scheme in condensed matter physics. In Sec. 3 we give details of the techniques used throughout this study: i) in the microcanonical ensemble, we use the microcanonical Lanczos (MCLM) method (Sec. 3.2.1), the microcanonical (MTPQ) variant of the TPQ (Sec. 3.2.2), and the iterative Chebyshev polynomial Green's function (CPGF) method (Sec. 3.2.3); ii) in the canonical ensemble, we use the finite temperature Lanczos (FTLM) method (Sec. 3.3.1), the canonical (CTPQ) variant of the TPQ (Sec. 3.3.2), and the newly developed finite temperature Chebyshev polynomial (FTCP) method that we introduce in this work (Sec. 3.3.3); iii) for dynamical studies, we use the Lanczos method using the continued fraction approach and a hybrid Lanczos-Chebyshev approach, also introduced in this work (Sec. 3.4.2). Section 4 compares the convergence properties and performance of all these methods in detail by applying them to study the K-H and K-I models on the honeycomb lattice: i) we start by checking consistency at zero temperature by computing the ground state energy and nearest-neighbor spin–spin correlation function of the K-H model with all methods and comparing them with previously known results that we reproduced using the ED Lanczos technique; ii) for the spin–spin correlation, we present a detailed analysis of the dependence of our estimators on the relevant parameters: the number of initial random states, truncation order and, in the case of the CPGF, also the energy resolution; iii) we study the temperature dependence of the nearest-neighbor spin–spin correlation, specific heat and entropy of the K-H model and show that our results match the tensor network results of Ref. [10]; iv) we bench-mark our implementation of the hybrid Lanczos-Chebyshev method and compute the dynamical spin susceptibility for the K-I model, finding dynamical signatures of the quantum phase transitions described in Ref. [61]. Finally, in Sec. 5, we point out the pros and cons of each method. In particular, we summarize how this work highlights the efficiency of the CPGF, FTCP and the hybrid Lanczos-Chebyshev methods and suggest potentially interesting applications of these methods. We also discuss the dynamical signatures of the quantum phase transitions in the K-I model that we found using the hybrid Lanczos-Chebyshev approach.

---

[1]The scheme in Refs. [70,71] requires the partial diagonalization of the original Hamiltonian in terms of Majorana fermions. On the other hand, a direct QMC simulation of the Kitaev model suffers from the sign problem.

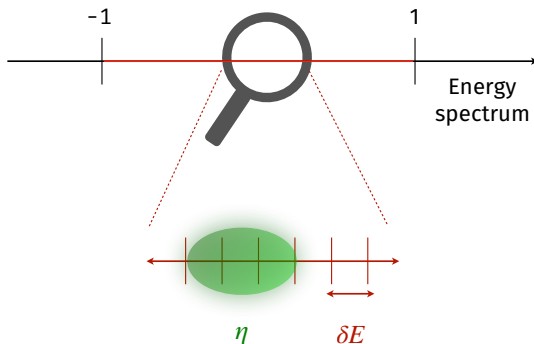

Figure 1: Microcanonical CPGF approach, the rescaled energy spectrum is probed using a coarse-grained average of energy states within a specified energy resolution $\eta$.

## 2   Chebyshev spectral methods: Rationale

Spectral methods are an increasingly popular tool for the simulation of condensed matter systems that fulfill the requirement of general applicability [49,52,56–60,72–77]. These methods rely on the iterative reconstruction of the target functions (e.g. static or dynamic correlation functions), generally in terms of Chebyshev polynomial expansions due to their favorable convergence properties [78]. The iterative scheme is stable and can be made as accurate as required within a specified parameter. For example, if one is interested in computing expectations of quantum observables in the microcanonical ensemble, this parameter is the energy resolution. The canonical ensemble analogue of this parameter is the temperature.

Let us take the instructive case of the microcanonical ensemble. The spectral approach uses a coarse-grained description of energy states to provide estimates for quantum observables in large systems (see Fig. 1). This is to be contrasted with exact diagonalization, which relies on the knowledge of individual states and thus is limited to small systems. Furthermore, spectral methods can be combined with stochastic techniques for computation of traces to further reduce the computational cost and are amenable to parallelization as we will see briefly.

The CPGF approach we exploit in this work to compute microcanonical averages [51, 79] has proven effective in dealing with tight-binding models, allowing unparalleled large-scale simulations with billions of atomic orbitals [51, 55]. Motivated by these developments, the main aim of this work is to introduce a finite-temperature spectral framework that can capture the physics of two-dimensional quantum spin models over a wide range of temperatures (of particular interest will be to probe the low-temperature behavior of spin liquids).

Spectral methods leverage efficient stochastic estimators for expectation values that use random vectors[2] to evaluate traces of operators [80]. This technique, dubbed stochastic trace evaluation (STE), is ubiquitous in the study of condensed phases and is used in ED methods, such as those based on the Lanczos algorithm and TPQ states [34,35], and in the kernel polynomial method [49]. The rationale in the STE is to approximate the trace of an operator by an average of expectation values using $N_{\text{rd.vec.}}$ random vectors, $|\phi_0^{(r)}\rangle$ — where $r$ is an index labelling a specific realization of the random vector $|\phi_0\rangle$ — i.e. $\text{Tr}_{\text{STE}}\,\hat{O} := \frac{1}{N_{\text{rd.vec.}}}\sum_{r=1}^{N_{\text{rd.vec.}}}\langle\phi_0^{(r)}|\hat{O}|\phi_0^{(r)}\rangle$. The relative error scales favorably with the Hilbert space dimension, $D$ (in fact the relative

---

[2]A random vector is defined as $|\phi_0\rangle = \sum_{i=1}^{D}\xi_i|i\rangle$, with $\{|i\rangle\}$ an arbitrary basis and $\xi_i \in \mathbb{C}$ random variables that satisfy $\overline{\xi_i} = 0$, $\overline{\xi_i\xi_j} = \delta_{ij}$ and $\overline{\xi_i^*\xi_j} = \delta_{ij}$ (here the bar denotes statistical average). The $r$-th realization, $|\phi_0^{(r)}\rangle$ corresponds to a specific set of coefficients, $\{\xi_i^r, i = 1, 2, \ldots, D.\}$ and different sets are assumed to be uncorrelated.

error is proportional to $1/\sqrt{D}$ for typical sparse operators) and, for a fixed system size, can be made as small as desired by increasing $N_{\text{rd.vec.}}$. Moreover, stochastic trace estimators are free from the sign problem.

A crucial feature of Chebyshev expansions is that they offer uniform convergence [78] (and in the case of CPGF this translates into an energy resolution that can be specified exactly [51]). This is appealing for studies of phase transitions, particularly when one wishes to characterize the critical behavior of thermodynamic functions, among others. There are two ways to define a resolution. One of them uses a kernel that modifies the coefficients of the Chebyshev expansion. This modification smears out so called Gibbs oscillations which occur upon truncation of orthogonal polynomial series [49]. A resolution may then be defined as the spread of the kernel in the $xy$-plane,[3] and generally depends on the truncation order and energy. Here, instead, we use a Green's function-based method that was proposed independently in the works of Ferreira and Mucciolo [51] and Braun and Schmitteckert [79]. This approach, coined CPGF [51], has two main features: (i) it is based on a stable, asymptotically exact expansion of lattice Green's functions in Chebyshev polynomials; and (ii) the energy resolution is specified from the outset, in the form of a simple imaginary self-energy.

By extending these ideas to quantum spin models, we propose a Chebyshev-based method for the computation of quantum expectations in the canonical ensemble, where the temperature plays the role of a resolution. This method — which we shall detail below — bypasses potential low-temperature convergence issues by using an adaptive temperature step, while maintaining rigorous control over convergence. Alternatively, TPQ-based methods can be used to approximate either microcanonical [34] or canonical [35] averages by successive application of the Hamiltonian operator onto an initial random state. In TPQ, the number of iterations is proportional to a quantity that plays the role of an effective temperature. Broadly speaking, this effective temperature acts similarly to a resolution that becomes finer as more iterations are completed. Yet, this annealing scheme is susceptible to slow convergence, particularly in the vicinity of critical points as shown later.

In the following sections, we will compare Lanczos, TPQ, and Chebyshev-based approaches since they all scale linearly with the dimension of the Hilbert space $D$. Moreover, all methods scale linearly with the number of polynomials required for spectral convergence (or iterations in the case of TPQ) $N_{\text{poly/it.}}$ and with the number of realizations of the initial random state required for statistical convergence $N_{\text{rd.vec.}}$.

## 3 Methodology

The methods described here involve two main steps. First, a numerically exact or approximate spectral representation of the target state is obtained by means of algorithms with polynomial computational complexity. This is achieved by recursive application of the Hamiltonian, $\hat{H}$, to an initial random state, $|\phi_0\rangle$. Some examples of typical target states are the ground state, a microcanonical state (with energy restricted to an energy shell) or a canonical (finite temperature) state. After the target state is converged to the desired precision, physical observables may be computed by means of the STE technique introduced earlier in Sec. 2, i.e. by averaging the expectation value $\langle\phi_0|\hat{O}|\phi_0\rangle$ over an ensemble of random vectors.

These techniques are general in scope and, as discussed below, when combined provide a powerful means of accessing excited states, reconstructing Green's functions, computing average and local density of states, and can be easily extended to the study of quantum dynamics, either in the time domain, by exploiting a spectral approximation of the time evolution operator, or in the frequency domain, via the resolvent operator.

---

[3]The two variables $x, y$ are defined as follows. $x$ is the variable upon which the function we wish to approximate depends on, say the energy. $y$ is the integration variable used when the function is convoluted with the kernel.

## 3.1  Lanczos exact diagonalization

While the iterative methods discussed here generate different polynomials of the Hamiltonian during the recursion, they have a crucial aspect in common. After $M$ iterations, they create a state in the so called Krylov subspace, defined as follows:

$$\mathcal{K}_M(\hat{h}, |\phi_0\rangle) \equiv \text{span}\{|\phi_0\rangle, \hat{h}|\phi_0\rangle, \hat{h}^2|\phi_0\rangle, \ldots \hat{h}^M|\phi_0\rangle\}, \tag{1}$$

where $\hat{h} = \hat{H}/N$ is the Hamiltonian normalized to the number of lattice sites (referred to as *Hamiltonian density* throughout) and $|\phi_0\rangle$ is a normalized initial random state. The Lanczos method [13] converges quickly to the ground state and low-lying excitations. It consists of iteratively generating a set of orthonormal states $\{|\phi_j\rangle, j = 0, 1, \ldots, M\}$ spanning the Krylov space. Let $\alpha_j = \langle\phi_j|\hat{h}|\phi_j\rangle$. Then, $\alpha_0$ is used to generate an unnormalized orthogonal state:

$$|\Phi_1\rangle = (\hat{h} - \alpha_0)|\phi_0\rangle, \tag{2}$$

which can then be normalized to obtain the second Lanczos state:

$$|\phi_1\rangle = \beta_1^{-1}|\Phi_1\rangle, \qquad \beta_1 = \sqrt{\langle\Phi_1|\Phi_1\rangle}. \tag{3}$$

Subsequent Lanczos states are generated using the recursion and normalization scheme:

$$|\Phi_{j+1}\rangle = (\hat{h} - \alpha_j)|\phi_j\rangle - \beta_j|\phi_{j-1}\rangle, \quad |\phi_{j+1}\rangle = \beta_{j+1}^{-1}|\Phi_{j+1}\rangle, \quad j = 1, 2, \ldots, M-1, \tag{4}$$

with $\beta_j = \sqrt{\langle\Phi_j|\Phi_j\rangle}$. Notice that acting with $\langle\Phi_1|$ upon Eq. (2), yields $\langle\phi_1|\hat{h}|\phi_0\rangle = \beta_1$. Other nonzero matrix elements of the Hamiltonian are obtained by acting with either $\langle\phi_{j-1}|$, $\langle\phi_j|$, or $\langle\phi_{j+1}|$ on the recursion of Eq. (4):

$$\langle\phi_{j-1}|\hat{h}|\phi_j\rangle = \beta_j, \quad \langle\phi_j|\hat{h}|\phi_j\rangle = \alpha_j, \quad \langle\phi_{j+1}|\hat{h}|\phi_j\rangle = \beta_{j+1}. \tag{5}$$

Thus, the representation of the Hamiltonian in the Lanczos basis is a tridiagonal matrix, which is exact when $M$ coincides with the size of the Hilbert space, $D$. A low-energy approximation of the Hamiltonian is obtained by truncating the tridiagonal matrix at $M \ll D$:

$$T_M = \begin{pmatrix} \alpha_0 & \beta_1 & 0 & \ldots & 0 \\ \beta_1 & \alpha_1 & \beta_2 & \ddots & \vdots \\ 0 & \beta_2 & \ddots & \ddots & 0 \\ \vdots & \ddots & \ddots & \ddots & \beta_M \\ 0 & \ldots & 0 & \beta_M & \alpha_M \end{pmatrix}. \tag{6}$$

In our Lanczos implementation, $T_M$ is diagonalized using the method of Multiple Relatively Robust Representations (MR) [81], implemented e.g. in LAPACK [82–84]. MR was chosen to maximize efficiency because it has $\mathcal{O}(M^2)$ computational complexity and allows one to specify a range of desired eigenpairs, rather than computing all eigenpairs. This is useful because we are only interested in the lowest eigenvalues of $T_M$, $\{\varepsilon_{j=0,1,\ldots,\lambda}\}$ (with $\lambda \ll M$), which accurately approximate the low-lying eigenvalues of $\hat{h}$. The corresponding eigenstates, $\{|\psi_j\rangle\}$ are obtained by transforming to the original basis using the eigenvectors of $T_M$, $\mathbf{v}_j = (v_{j0}, v_{j1}, \ldots, v_{jM})$:

$$|\psi_j\rangle = \sum_{i=0}^{M} v_{ji}|\phi_i\rangle. \tag{7}$$

The dominant memory cost of the methods discussed throughout is incurred via the storage of vectors of dimension $D$ ($D$-vectors). This is because the Hamiltonian is never stored in

memory, e.g. as a sparse matrix. Instead, the matrix-vector multiplications encoding the action of the Hamiltonian on a state are carried out "on-the-fly", based on the bit representation of spin states explained in Ref. [85]. For completeness, we provide a brief description. Each of the $D = 2^N$ states in a basis of product states of individual spins is encoded by an integer between 0 and $D-1$, represented by a set of bits. Mapping the lattice sites to the $N$ bit positions and the individual spin states to the value of the bit (either 0 or 1), the Hamiltonian acts on a basis state in one of the two ways. Either the state: i) gets multiplied by a constant that depends on the value of two bits at different positions; or ii) it gets converted to a state encoded by a different integer, obtained by flipping only two bits, and then multiplied by a constant that depends on the values of the two flipped bits. Once a model is translated into these simple rules — which are stored at virtually no memory cost — any matrix-vector multiplication boils down to applying the rules to basis states. In particular, the Lanczos recursion requires only two $D$-vectors ($|\phi_i\rangle, |\phi_{i-1}\rangle$) to be stored in memory in each step, $i$. Consequently, constructing the corresponding eigenstates, $|\psi_j\rangle$ entails a second Lanczos recursion in order to regenerate the Lanczos vectors, while accumulating the weighted sum of Eq. (7). This can only be done once the eigenvectors $\mathbf{v}_j$ are obtained at the end of the first recursion. Suppose we are interested in constructing one of the low-lying states, $|\psi_j\rangle$. Then, we require an additional vector to be stored in memory so as to accumulate the weighted sum of Eq. (7) during the second recursion, implying that the memory cost is dominated by three $D$-vectors.

Once a low-lying eigenstate, $|\psi_j\rangle$, is found, the static expectation value of a quantum observable in that state can be evaluated. If multiple low-lying states are desired, it is still possible to preserve the 3-vector memory cost by carrying out multiple Lanczos recursions to evaluate the relevant expectations for different low-lying states. In contrast, constructing the whole set of eigenstates, $\{|\psi_j\rangle\}$ during the second recursion requires as many extra vectors as desired eigenstates to be stored in memory. While the ground state and low-lying excitations are important, there are problems that require knowledge of higher-excited states or even the whole spectrum. Below, we compare different approaches to go beyond low-lying states.

## 3.2 Microcanonical ensemble

In the standard Lanczos algorithm, the core of the spectrum of $\hat{h}$ is inaccessible. Loss of orthogonality due to finite machine precision impedes convergence beyond low-lying excitations. To complicate matters further, reorthogonalization schemes are computationally expensive [85]. An alternative approach is to set a target energy, construct a quasi-eigenstate corresponding to that energy and compute observables using the obtained microcanonical state.

### 3.2.1 Microcanonical Lanczos

In the microcanonical Lanczos method (MCLM) [17, 86], excited states in the core of the spectrum — and thus inaccessible to the "ground state" Lanczos method above — are probed by setting a target energy density, $\varepsilon$ and finding the lowest-lying eigenpair of

$$\hat{v} = (\hat{h} - \varepsilon)^2 \,. \tag{8}$$

The lowest eigenvalue found by performing a Lanczos recursion with $\hat{v}$ approaches 0 and, using its corresponding Lanczos vectors, one can construct the quasi-eigenstate $|\psi_\varepsilon\rangle$. The MCLM converges slower than the standard Lanczos approach in most applications. For example, the microcanonical variant was found to require $\mathcal{O}(10^3)$ iterations to construct quasi-eigenstates in the spin-1/2 Heisenberg chain [86]. This is to be contrasted to the standard Lanczos algorithm, which typically requires $\mathcal{O}(10^2)$ iterations to retrive a low-lying state [17]. The energy uncertainty reads

$$\sigma_\varepsilon = \sqrt{\langle \psi_\varepsilon | \hat{v} | \psi_\varepsilon \rangle} \,. \tag{9}$$

As a rule of thumb, in Ref. [17], the authors state that in order to resolve the desired energy level with small energy spread $\sigma_\varepsilon/W < 10^{-3}$, where $W$ is the spectrum width, $M' \sim 10^3$ iterations are typically needed. Then, the quasi-eigenstate can be used to compute observables reliably. The computational complexity is dominated by two main components. One of them comes from the diagonalizations of the tridiagonal matrices at each Lanczos iteration. Since we use MR [81, 83, 84] for these diagonalizations, the number of floating point operations for this part scale as

$$\sum_{m=1}^{M'} m^2 = M'(M'+1)(2M'+1)/6 \sim \mathcal{O}(M'^3). \tag{10}$$

The other component comes from the $M'$ matrix-vector multiplications in the Lanczos recursion, each carried out "on-the-fly", incurring a cost $\mathcal{O}(zD\log_2 D)$, where $z$ is the coordination number of the lattice. Thus, the computational effort from matrix-vector multiplications scales as $\mathcal{O}(zM'D\log_2 D)$, where $D = 2^N$ for spin-1/2 systems.

For $N \gtrsim 20$, the ratio between the two costs is $zD\log_2 D/M'^2 \gtrsim 1$, and the computational complexity is dominated by the cost of matrix-vector multiplication. However, if the more standard implicit QR method is used for diagonalization instead of MR, the complexity of the diagonalization increases to $\mathcal{O}(M'^4)$. The relevant ratio of computational costs becomes $zD\log_2 D/M'^3$, which only becomes significantly larger than 1 for $N \gtrsim 30$.

As a final note on memory cost, we remark that the $\hat{h}^2$-term in Eq. (8) requires an additional vector to be stored in memory compared with the "ground state" Lanczos, increasing the number of stored $D$-vectors to four.

### 3.2.2 Thermal pure quantum states

In this subsection, we follow closely the work of Sugiura and Shimizu [34]. The rationale of the TPQ method is to find a pure state that faithfully captures the equilibrium properties of a quantum system at finite temperature as accurately as possible using microcanonical TPQ states with well defined energy, constructed as follows. First, one generates a random state

$$|\phi_0\rangle \equiv \sum_{i=1}^{D} \xi_i |i\rangle . \tag{11}$$

For simplicity, $\{|i\rangle\}$ is usually taken as the set of product states of individual spins. The distribution of energy in $|\phi_0\rangle$ is proportional to the density of states

$$g(u;N) = \exp[Ns(u;N)], \tag{12}$$

where $s(u;N)$ is the entropy density, which converges to a function of the energy density, $s(u;\infty)$, in the thermodynamic limit [34].

The basic procedure is an iterative one similar to minimization annealing schemes. The goal is to modify the distribution of energy in the random state so that it becomes sharply peaked at the desired energy density, $\varepsilon$. This is achieved by operating with a suitable polynomial of the Hamiltonian density onto $|\phi_0\rangle$ iteratively. Take a constant $\varepsilon_{\text{upper}} \sim \mathcal{O}(1)$, such that $\varepsilon_{\text{upper}} \geq \varepsilon_{\text{M}}$, where $\varepsilon_{\text{M}}$ is the maximum eigenvalue of the Hamiltonian density. Thus, $\varepsilon_{\text{upper}}$ is an upper bound on the spectrum of $\hat{h}$. Then, start from $|\phi_0\rangle$ and iteratively compute, respectively the energy density, $u_k$ and the (normalized) new state, $|\phi_{k+1}\rangle$ at iteration $k$:

$$u_k = \langle\phi_k|\hat{h}|\phi_k\rangle ,$$
$$|\phi_{k+1}\rangle = \frac{|\Phi_{k+1}\rangle}{\sqrt{\langle\Phi_{k+1}|\Phi_{k+1}\rangle}}, \quad \text{where} \quad |\Phi_{k+1}\rangle \equiv (\varepsilon_{\text{upper}} - \hat{h})|\phi_k\rangle , \tag{13}$$

iteratively for $k = 1, 2, \ldots, N_{it}$, the maximum number of iterations. Later on, we will see that $N_{it}$ plays a role analogous to the inverse of the resolution in the Chebyshev expansion. Since the microcanonical states are now generated directly, as opposed to being reconstructed via a Lanczos recursion, the memory cost is now dominated by two $D$-vectors rather than four.

The first energy density corresponds to the infinite temperature state at $\beta = 0$. Thus, $g(u; N)$ has its maximum at $u = u_0$. Then, the energy density decreases gradually towards the ground state energy, $\varepsilon_m$, as $k$ is increased: $u_0 > u_1 > \cdots > u_{N_{it}} \geq \varepsilon_m$. One stops iterating at $k = N_{it}$, when $u_k$ gets close enough to the ground state energy density, $\varepsilon_m$. The obtained TPQ states in the sequence $|\phi_0\rangle, |\phi_1\rangle, \ldots, |\phi_{N_{it}}\rangle$ correspond to decreasing thermal energy densities $u_0 > u_1 > \cdots > u_{N_{it}}$. Thus, an estimate of the equilibrium average value of an arbitrary observable $\hat{A}$ is obtained as $\langle \hat{A} \rangle_k = \langle \phi_k | \hat{A} | \phi_k \rangle$ as a function of $u_k$. Notably, in the large system limit, the effective temperature associated with each TPQ iteration, $\beta_k$, accurately reproduces the true thermodynamic temperature of a state with energy density $u_k$. In fact, it is possible to approximate the thermodynamic temperature with an error of order $\mathcal{O}(1/N^2)$ [34].

Finally, the static expectation value $\langle \hat{A} \rangle_k$ obtained for each realization of the random coefficients $\{\xi_i\}$ depends exponentially less on the number of sites, $N$, as the latter is increased, due to self averaging properties. Hence, accurate results are often obtained with a few or even a single random vector realization [34, 35].

### 3.2.3 Chebyshev polynomial Green's function

This method consists of numerically evaluating the lattice resolvent operator

$$\hat{G}(z) = (z - \hat{h})^{-1}, \tag{14}$$

via an exact expansion in terms of Chebyshev polynomials of the Hamiltonian density. Here, $z = \varepsilon + i\eta$ is a complex energy variable. A key aspect is that the Green's function is reconstructed with uniform energy resolution over the entire energy range. For numerical stability, the resolution parameter should satisfy $\eta = \mathrm{Im}\, z \gtrsim \delta\varepsilon$, where $\delta\varepsilon$ is the mean level spacing. To expand Eq. (14) in Chebyshev polynomials, we consider the following linear transformation of the Hamiltonian and the energy variables: $\tilde{h} = (\hat{h} - b)/a$ and $\tilde{z} = \tilde{\varepsilon} + i\tilde{\eta}$, where $\tilde{\varepsilon} = (\varepsilon - b)/a$, $\tilde{\eta} = \eta/a$, and

$$a = f \frac{\varepsilon_M - \varepsilon_m}{2}, \qquad b = \frac{\varepsilon_M + \varepsilon_m}{2}, \tag{15}$$

where $\varepsilon_M$ and $\varepsilon_m$ are the extremal eigenvalues and $f \simeq 1.001$ is a safety factor to ensure that the spectrum of the reconstructed operator falls inside the Chebyshev domain of convergence at each iteration step. As customary, we work with Chebyshev polynomials of the first kind $\{T_n(x) = \cos(n \arccos x), n = 0, 1, 2 \ldots\}$ due to their favorable convergence properties [78].

Typical target functions of energy, including density of states and static expectations values, are evaluated by making use of the Chebyshev polynomial expansion of the imaginary part of the rescaled Green's function [51]

$$\mathrm{Im}[\hat{G}(\tilde{\varepsilon} + i\tilde{\eta})] = \sum_k \frac{\tilde{\eta}}{(\tilde{\varepsilon} - \tilde{\varepsilon}_k)^2 + \tilde{\eta}^2} |k\rangle \langle k| = \sum_{n=0}^{\infty} \mathrm{Im}[g_n(\tilde{z})] T_n(\tilde{h}), \tag{16}$$

with

$$g_n(z) = \frac{-2i}{1 + \delta_{0,n}} \frac{(z - i\sqrt{1 - z^2})^n}{\pi \sqrt{1 - z^2}}. \tag{17}$$

The operators $T_n(\tilde{h})$ of Eq. (16) are constructed using the operator versions of the Cheby-

shev recursion relation

$$
\begin{aligned}
T_0(\tilde{h}) &= 1\,, \\
T_1(\tilde{h}) &= \tilde{h}\,, \\
T_{n+1}(\tilde{h}) &= 2\tilde{h}T_n(\tilde{h}) - T_{n-1}(\tilde{h})\,.
\end{aligned}
\tag{18}
$$

The series is truncated when the desired accuracy is achieved for a given choice of resolution. The $(N_{\text{poly}} + 1)$-th order approximation of the lattice Green's function is therefore

$$
\hat{G}_{N_{\text{poly}+1}}(\tilde{\varepsilon} + i\tilde{\eta}) \equiv \sum_{n=0}^{N_{\text{poly}}} g_n(\tilde{\varepsilon} + i\tilde{\eta})T_n(\tilde{h})\,.
\tag{19}
$$

For most cases, $N_{\text{poly}} = c\tilde{\eta}^{-1}$, with $c = \mathcal{O}(1)$ sufficing to achieve machine precision [55]. The spectral operator within the CPGF approach can be defined as follows

$$
\delta_{\tilde{\eta}}(\tilde{\varepsilon} - \tilde{h}) = -\sum_n \text{Im}[g_n(\tilde{\varepsilon} + i\tilde{\eta})]T_n(\tilde{h})\,.
\tag{20}
$$

By applying this operator to the $r$-th realization of the random state $|\phi_0^{(r)}\rangle$, we obtain $|\tilde{\varepsilon}, \tilde{\eta}\rangle_r$, a quasi-eigenstate with rescaled energy $\tilde{\varepsilon}$ (within the rescaled resolution $\tilde{\eta}$). To compute static expectation values, we start by defining

$$
\{\hat{A}\}_r(\varepsilon, \eta) \equiv \langle \varepsilon, \eta | \hat{A} | \varepsilon, \eta \rangle_r = a^{-2}\langle \tilde{\varepsilon}, \tilde{\eta} | \hat{A} | \tilde{\varepsilon}, \tilde{\eta} \rangle_r\,.
\tag{21}
$$

Provided that the resolution $\eta$ is adequate ($\eta \to \delta\varepsilon \Leftrightarrow \tilde{\eta} \to \delta\varepsilon/a$, where $\delta\varepsilon$ is the mean level spacing), one obtains an accurate estimate of the expectation value of $\hat{A}$ for a given energy $\varepsilon$, $A(\varepsilon)$ by averaging over realizations of the initial random state and using Eq. (21):

$$
\langle A \rangle_{\text{STE}}(\varepsilon, \eta) = \frac{\sum_{r=1}^{N_{\text{rd.vec.}}} \{\hat{A}\}_r(\varepsilon, \eta)}{\sum_{r=1}^{N_{\text{rd.vec.}}} \{\hat{1}\}_r(\varepsilon, \eta)} \xrightarrow[\eta \to \delta\varepsilon^+]{} A(\varepsilon)\,.
\tag{22}
$$

### 3.3 Canonical ensemble

While microcanonical methods are useful, one might also be interested in evaluating observables using canonical states. In practice, one may decide which ensemble is more convenient to perform a given calculation because the principle of ensemble equivalence guarantees that results are consistent across statistical ensembles. For example, canonical methods have the advantage of allowing direct specification of temperature as an input, so they may be preferable to study temperature dependence of systems in thermal equilibrium. Moreover, for finite systems, calculations done with the microcanonical ensemble tend to show significant statistical fluctuations [17]. As temperature increases, higher energy states in the spectrum become increasingly important for determining the properties of the system and these statistical fluctuations are smeared out. The most interesting features of the systems we tackle in this work appear at low temperature, so the canonical methods detailed below are particularly useful in this context.

#### 3.3.1 Finite temperature Lanczos method

The finite temperature Lanczos method (FTLM) has been introduced in Ref. [15] and discussed in depth in Ref. [17]. The basic idea is to generate a set of eigenpairs $\{\varepsilon_{j,r}, |\psi_j^{(r)}\rangle\}$ using $M_{\text{FT}}$ Lanczos steps and starting from different realizations of the initial random state. Throughout the recursion, we need only store two sets of overlaps: $Q_{r,j} \equiv \langle \phi_0^{(r)} | \psi_j^{(r)} \rangle$ and

$A_{r,j} \equiv \langle \psi_j^{(r)} | \hat{A} | \phi_0^{(r)} \rangle$. The memory cost is still dominated by the $D$-vectors: two for the recursion and one to store the initial random state so as to allow the computation of the overlaps, totaling three $D$-vectors. The STE estimator of the canonical average, $\langle A \rangle (\beta, N)$, defined as

$$\langle A \rangle (\beta, N) \equiv \frac{\text{Tr}[\hat{A} e^{-\beta \hat{H}}]}{\text{Tr}[e^{-\beta \hat{H}}]} = \frac{\text{Tr}[e^{-\beta \hat{H}/2} \hat{A} e^{-\beta \hat{H}/2}]}{\text{Tr}[e^{-\beta \hat{H}}]}, \tag{23}$$

in the FTLM [15,17] is obtained as

$$\langle A \rangle_{\text{STE}} (\beta, N) = \frac{\sum_{r=1}^{N_{\text{rd.vec.}}} \sum_{j=0}^{M_{\text{FT}}} e^{-N\beta \varepsilon_{j,r}} Q_{r,j} A_{r,j}}{\sum_{r=1}^{N_{\text{rd.vec.}}} \sum_{j=0}^{M_{\text{FT}}} e^{-N\beta \varepsilon_{j,r}} |Q_{r,j}|^2}. \tag{24}$$

Since the statistical fluctuations of this estimator increase significantly as the temperature is decreased, the need for an optimized low-temperature Lanczos method (LTLM) arose and this method has been introduced in Ref. [16]. Apart from the $Q_{r,j}$ overlaps defined above, the computation of the STE estimator for this method requires the storage of $\mathcal{O}(M_{\text{FT}}^2)$ additional overlaps: $A'_{r,l,j} \equiv \langle \psi_j^{(r)} | \hat{A} | \psi_l^{(r)} \rangle$, with $l, j = 0, 1, \ldots, M_{\text{FT}}$. In terms of these overlaps and using the symmetric form in the right hand side of Eq. (23), the final estimator becomes

$$\langle A \rangle_{\text{STE}} (\beta, N) = \frac{\sum_{r=1}^{N_{\text{rd.vec.}}} \sum_{l,j=0}^{M_{\text{FT}}} e^{-N\beta (\varepsilon_{l,r} + \varepsilon_{j,r})/2} Q_{r,l} A'_{r,l,j} Q_{r,j}^\star}{\sum_{r=1}^{N_{\text{rd.vec.}}} \sum_{j=0}^{M_{\text{FT}}} e^{-N\beta \varepsilon_{j,r}} |Q_{r,j}|^2}. \tag{25}$$

Notice that the LTLM requires a double sum with $\mathcal{O}(M_{\text{FT}}^2)$ terms. It becomes increasingly more expensive to compute these overlaps as the temperature is decreased and more Lanczos iterations $M_{\text{FT}}$ are required. However, the estimator of Eq. (25) has the advantage of reaching smoothly the zero temperature limit — as opposed to that of Eq. (24) — which is the reason behind its advantageous statistical convergence properties. Below, we introduce other alternative methods to FTLM since LTLM has an inherently high cost.

### 3.3.2 Canonical thermal pure quantum states

The microcanonical TPQ state we outlined previously is specified by the independent variables $(u, N)$. It can be shown [35] that its (unnormalised) canonical counterpart $|\beta, N\rangle$ — specified by the inverse temperature, $\beta$ instead of $u$ — is obtained as follows:

$$|\beta, N\rangle \equiv e^{-N\beta \hat{h}/2} |\phi_0\rangle. \tag{26}$$

A simple analytic transformation reminiscent of the principle of ensemble equivalence allows one to cast canonical TPQ states in terms of their microcanonical counterparts. This correspondence is obtained as follows. First, we assume that the minimum and maximum eigenvalues of $\hat{h}$, respectively $\varepsilon_m$ and $\varepsilon_M$, are known. These can be obtained numerically, for example with Lanczos. Let us now define an unnormalised microcanonical TPQ state for a given realization of the initial random state:

$$|k^{(r)}\rangle = \left( \frac{\varepsilon_M - \hat{h}}{W} \right)^k |\phi_0^{(r)}\rangle, \tag{27}$$

where $W = \varepsilon_M - \varepsilon_m$ is the bandwidth. Here, dividing by $W$ ensures some degree of numerical stability since the operator inside parentheses is then bounded. Multiplying and dividing Eq. (26) by $e^{N\beta \varepsilon_M/2}$ and Taylor expanding the exponential, one finds:

$$|\beta, N\rangle = e^{-N\beta \varepsilon_M/2} \sum_{k=0}^{\infty} \frac{(N\beta W/2)^k}{k!} |k\rangle. \tag{28}$$

In the canonical TPQ formulation (CTPQ), the STE estimator is then given by

$$\langle A \rangle_{\text{STE}}(\beta, N) = \frac{\sum_{r=1}^{N_{\text{rd.vec.}}} \langle \beta, N | \hat{A} | \beta, N \rangle_r}{\sum_{r=1}^{N_{\text{rd.vec.}}} \langle \beta, N | \beta, N \rangle_r}, \tag{29}$$

where the subscript $r$ means that the canonical states of Eq. (28) have been constructed using the $r$-th realization of the initial random state. Naively, one might expect that computing this expectation would involve performing a double sum over the iterations and storing $\mathcal{O}(N_{\text{it}}^2)$ overlaps, yielding a cost comparable to LTLM. This is because Eq.(28) implies that

$$\langle \beta, N | \hat{A} | \beta, N \rangle_r \propto \sum_{k,q} \frac{(N\beta W/2)^{k+q}}{k!q!} \langle k | \hat{A} | q \rangle_r, \quad \text{with} \quad \langle k | \hat{A} | q \rangle_r = \langle k^{(r)} | \hat{A} | q^{(r)} \rangle. \tag{30}$$

If the observable of interest is a constant of motion (i.e. $[\hat{A}, \hat{h}] = 0$), Eq.(30) simplifies significantly. Let $A_{k,r} = \langle k | \hat{A} | k \rangle_r$, $A'_{k,r} = \langle k | \hat{A} | k+1 \rangle_r$. Then, we have

$$\langle \beta, N | \hat{A} | \beta, N \rangle_r \propto \sum_k \left[ \frac{(N\beta W/2)^{2k}}{(2k)!} A_{k,r} + \frac{(N\beta W/2)^{2k+1}}{(2k+1)!} A'_{k,r} \right] \equiv \{\hat{A}\}_r(\beta, N). \tag{31}$$

In such cases, we only need to store $4N_{\text{it}} \ll D$ overlaps for each random vector:

$$A_{k,r} = \langle k | \hat{A} | k \rangle_r, \qquad A'_{k,r} = \langle k | \hat{A} | k+1 \rangle_r, \qquad N_{k,r} = \langle k | k \rangle_r, \qquad N'_{k,r} = \langle k | k+1 \rangle_r.$$

Finally, for each inverse temperature, the STE expectation of Eq.(29) can be reconstructed using the stored overlaps:

$$\langle A \rangle_{\text{STE}}(\beta, N) = \frac{\sum_{r=1}^{N_{\text{rd.vec.}}} \{\hat{A}\}_r(\beta, N)}{\sum_{r=1}^{N_{\text{rd.vec.}}} \{\hat{1}\}_r(\beta, N)}. \tag{32}$$

While this derivation is only strictly valid when $[\hat{A}, \hat{h}] = 0$, in Ref. [35], the authors show that it holds remarkably well in general. We shall confirm this in practice below.

Memory-wise, CTPQ is similar to its microcanonical counterpart, requiring two $D$-vectors. Unfortunately, numerical instabilities build up rapidly as $k$ increases. Lower-temperature properties are thus challenging to probe.

### 3.3.3 Finite temperature Chebyshev polynomial approach

So far, we have reviewed the generalization of the ideas behind TPQ and Lanczos to the study of canonical expectations. In what follows, we introduce the finite temperature Chebyshev polynomial (FTCP) method, a new approach that we developed to extend the CPGF method to the canonical ensemble description of interacting quantum systems.

In Ref. [51], the $g$-coefficients of Eq. (16) are obtained by exploiting the operator version of the Jacobi-Anger identity

$$e^{-iz\tilde{h}} = \sum_{n=0}^{\infty} \frac{2i^{-n}}{1 + \delta_{n,0}} J_n(z) T_n(\tilde{h}), \tag{33}$$

where $J_n(z)$ is the Bessel function of order $n$. We follow a similar route, and seek a Chebyshev expansion of the operator $e^{-\beta\tilde{h}/2}$. Suppose we are interested in low-temperature behavior, i.e. a high inverse temperature, $\beta_{\text{max}}$. We could expand the operator $e^{-\beta_{\text{max}}\tilde{h}/2}$ in Chebyshev

polynomials directly as done e.g. in Ref. [87]. However, such an expansion is vulnerable to numerical instabilities for large $\beta_{\max}$ due to the rapid growth of the Bessel functions. Using

$$e^{-\beta_{\max}\tilde{h}/2} = \prod_{k=1}^{L} e^{-\delta\beta_k\tilde{h}}, \tag{34}$$

where $\sum_{k=1}^{L}\delta\beta_k = \beta_{\max}/2$, we can decompose $e^{-\beta_{\max}\tilde{h}/2}$ into a string of $L$ operators. This expansion enables us to bypass the divergent behavior of $J_n(z)$ in Eq. (33) for large negative imaginary arguments $z = -i\beta_{\max}$. Moreover, the inverse temperature steps, $\delta\beta_k$ need not be uniform, but may instead vary for each operator, $e^{-\delta\beta_k\tilde{h}}$, in our string of $L$ operators. This opens the door to the use of an adaptive temperature step.

Using the modified Bessel functions — which obey the relation $I_n(x) = i^{-n}J_n(ix)$, $x \in \mathbb{R}$ — we can use the Jacobi-Anger identity to cast each operator in our string of operators as a Chebyshev series:

$$e^{-\delta\beta_k\tilde{h}} = \sum_{n=0}^{\infty} \frac{2}{1+\delta_{n,0}} I_n(-\delta\beta_k)T_n(\tilde{h}). \tag{35}$$

Applying Eq. (35) to a random state $|\phi_0^{(r)}\rangle$ produces a sequence of approximate finite temperature states (with inverse temperatures $\beta_l = 2\sum_{k=1}^{l}\delta\beta_k$ with $l = 1, \dots, L$), with accuracy controlled by the truncation order. In practice, a Chebyshev truncation order $N_{\text{poly},k} \sim \mathcal{O}(10)$ ensures convergence for a typical inverse temperature step of $\delta\beta \lesssim 10^2$ (in rescaled units). The $l$-th finite temperature state reads

$$|\phi_l^{(r)}\rangle \equiv \sum_{n=0}^{N_{\text{poly},l}} \frac{2}{1+\delta_{0,n}} I_n(-\delta\beta_l)T_n(\tilde{h})|\phi_{l-1}^{(r)}\rangle, \tag{36}$$

where, as customary, the Chebyshev vectors are generated starting from the $r$-th realization of the initial random state. We note that, at all steps, the arguments of the fast growing modified Bessel functions need to be kept small in order to avoid numerical instabilities.

As the $l$-th operator in the string of operators is applied to $|\phi_{l-1}^{(r)}\rangle$, the canonical average of a quantum observable, $\hat{A}$, can be evaluated for the $l$-th inverse temperature. In order to compute a thermal average, it suffices to notice that

$$\langle\phi_0^{(r)}|e^{-\beta_l\tilde{h}/2}\hat{A}e^{-\beta_l\tilde{h}/2}|\phi_0^{(r)}\rangle = \langle\phi_l^{(r)}|\hat{A}|\phi_l^{(r)}\rangle. \tag{37}$$

Then, using the RHS of Eq. (23), we obtain the STE expectation with the FTCP method:

$$\langle A\rangle_{\text{STE}}(\beta_l, N) = \frac{\sum_{r=1}^{N_{\text{rd.vec.}}}\langle\phi_l^{(r)}|\hat{A}|\phi_l^{(r)}\rangle}{\sum_{r=1}^{N_{\text{rd.vec.}}}\langle\phi_l^{(r)}|\phi_l^{(r)}\rangle}, \quad \text{where} \quad 2\sum_{k=1}^{l}\delta\beta_k = \beta_l \leq \beta_{\max}. \tag{38}$$

The adaptive inverse temperature step used in this work allows us to maximize efficiency by focusing the computational effort at low temperature, where a finer temperature grid (and thus a larger spacing $\delta\beta_k$) is required to capture the key features of the systems at play. The reconstruction of $\langle A\rangle_{\text{STE}}(\beta_l, N)$ for a discrete set of $L$ inverse temperatures, $\{\beta_l, l = 1, 2, \dots, L\}$ involves storing $2L$ overlaps, $\langle\phi_l^{(r)}|\hat{A}|\phi_l^{(r)}\rangle$ and $\langle\phi_l^{(r)}|\phi_l^{(r)}\rangle$ for each random vector realization. The total number of Chebyshev iterations is thus $N_{\text{Cheb}} = \sum_{l=1}^{L} N_{\text{poly},1} \sim 10^3$. As shown shortly in Sec. 4, the FTCP favorable convergence properties will allow us to reach very low temperatures that are hard to access with FTLM and CTPQ. Finally, FTCP has the same memory requirement of three $D$-vectors as CPGF: two for the Chebyshev recursion and one to cumulatively generate the finite temperature state at each inverse temperature.

## 3.4 Dynamical properties

The prototype simulation aimed at studying the dynamics of a quantum system starts from a well defined initial state $|\Psi(t=0)\rangle$, such as the ground state of the model Hamiltonian at play, $|\text{GS}\rangle$, which can be obtained, e.g. using the Lanczos method. This initial state is then evolved using the time evolution operator. Successive small time steps are taken in order to maintain enough numerical accuracy, while keeping track of the evolution of quantities of interest, such as time-domain correlators of the type

$$G^{\hat{B}\hat{A}}(t) = \langle \text{GS}|\hat{B}(t)\hat{A}(0)|\text{GS}\rangle\,, \tag{39}$$

where $\hat{A},\hat{B}$ are two generic quantum observables in the Heisenberg picture. Both Lanczos and a Chebyshev-based approaches exist to approximate the time evolution operator. Within the Lanczos approach, the time evolution operator for a short time step $\delta t$ is approximated as

$$e^{-iN\delta t\hat{h}} \approx \sum_{j=0}^{M_t} e^{-iN\varepsilon_j\delta t}|\psi_j\rangle\langle\psi_j|\,, \tag{40}$$

where $\{\varepsilon_j\}$ and $\{|\psi_j\rangle\}$ are sets of energies and corresponding eigenstates obtained using $M_t$ Lanczos steps and starting the Lanczos procedure from a previously computed state $|\Psi(t')\rangle$. On the other hand, the Chebyshev approximation of the time evolution operator — which is used e.g. in Ref. [88] in combination with CTPQ[4] — relies on Eq.(33):

$$e^{-iN\delta t\tilde{h}} \approx \sum_{n=0}^{N_t} \frac{2i^{-n}}{1+\delta_{n,0}}J_n(N\delta t)T_n(\tilde{h})\,. \tag{41}$$

Both methods require $\mathcal{O}(10)$ iterations for a standard time step $\delta t \approx W^{-1}$ [17]. Yet, the Chebyshev approach has an important advantage. Unlike Lanczos, where a tridiagonal matrix has to be diagonalized at each time step to generate the coefficients of the Lanczos expansion, the coefficients in the Chebyshev expansion in Eq. (41) can be easily and efficiently evaluated using freely available numerical libraries.

The efficiency of the Chebyshev approximation of the time evolution operator suggests that a Chebyshev approach can also be advantageous when studying zero-temperature spectral functions, $\mathcal{C}^{\hat{B}\hat{A}}(\omega)$, obtained by Fourier transforming the time-domain correlators of Eq. (39):

$$\begin{aligned}
\mathcal{C}^{\hat{B}\hat{A}}(\omega) &= \int \frac{dt}{2\pi}e^{i\omega t}G^{\hat{B}\hat{A}}(t) \\
&= \langle\text{GS}|\hat{B}\delta(\omega-\hat{h}+\varepsilon_m)\hat{A}|\text{GS}\rangle \\
&= -\frac{1}{\pi}\lim_{\eta\to 0}\langle\text{GS}|\hat{B}\text{Im}\left(\frac{1}{\omega-\hat{h}+\varepsilon_m+i\eta}\right)\hat{A}|\text{GS}\rangle\,,
\end{aligned} \tag{42}$$

where we recall that $\varepsilon_m$ is the ground state energy density, obtained e.g. with Lanczos.

### 3.4.1 Dynamical autocorrelation response functions with Lanczos

In the particular case where $\hat{B} = \hat{A}^\dagger$, the spectral function $\mathcal{C}^{\hat{A}^\dagger\hat{A}}$ is referred to as the autocorrelation response function for observable $\hat{A}$ and is defined as follows (with $z = \omega + \varepsilon_m + i\eta$):

$$\mathcal{A}(\omega) = -\frac{1}{\pi}\lim_{\text{Im}\,z\to 0}\langle\text{GS}|\hat{A}^\dagger\text{Im}\big[(z-\hat{h})^{-1}\big]\hat{A}|\text{GS}\rangle\,. \tag{43}$$

---

[4]In Ref. [88], CTPQ is used to generate an initial thermal state at $t = 0$. Time evolution is carried out using the Chebyshev approximation of $e^{-it\hat{H}}$.

Once the ground state, $|\text{GS}\rangle$, is reconstructed with Lanczos, the response function above can be computed by performing an additional Lanczos recursion (where the number of iterations needed for satisfactory convergence is typically $\tilde{M} \sim 10^3$) with the initial state

$$|\tilde{\phi}_0\rangle = \frac{\hat{A}|\text{GS}\rangle}{\sqrt{\langle\text{GS}|\hat{A}^\dagger\hat{A}|\text{GS}\rangle}}. \tag{44}$$

Similarly to Sec. 3.1, this recursion also generates a (much larger) tridiagonal matrix

$$\tilde{T}_{\tilde{M}} = \begin{pmatrix} \tilde{\alpha}_0 & \tilde{\beta}_1 & 0 & \dots & 0 \\ \tilde{\beta}_1 & \tilde{\alpha}_1 & \tilde{\beta}_2 & \ddots & \vdots \\ 0 & \tilde{\beta}_2 & \ddots & \ddots & 0 \\ \vdots & \ddots & \ddots & \ddots & \tilde{\beta}_{\tilde{M}} \\ 0 & \dots & 0 & \tilde{\beta}_{\tilde{M}} & \tilde{\alpha}_{\tilde{M}} \end{pmatrix}, \tag{45}$$

whose entries can be used to compute the response function [17] with no need to compute the eigenpairs $\{\varepsilon_j, \tilde{\mathbf{v}}_j, \ j = 0, 1, \dots, \tilde{M}\}$. The resolvent $(z - \hat{h})^{-1}$ can be approximated using a continued fraction, thus giving the "Lanczos" response function

$$\mathcal{A}(\omega, \eta) = -\frac{1}{\pi}\text{Im} \cfrac{\langle\text{GS}|\hat{A}^\dagger\hat{A}|\text{GS}\rangle}{\omega + \varepsilon_m + i\eta - \tilde{\alpha}_0 - \cfrac{\tilde{\beta}_1^2}{\omega + \varepsilon_m + i\eta - \tilde{\alpha}_1 - \cfrac{\tilde{\beta}_2^2}{\omega + \varepsilon_m + i\eta - \dots}}}, \tag{46}$$

where the continued fraction is terminated with $\tilde{\beta}_{\tilde{M}+1} = 0$. This procedure is signficantly more expensive computationally than simply approximating the ground state with Lanczos as described in Sec. 3.1. This is due to the accumulated cost of matrix-vector multiplications as more iterations are completed, which is $\mathcal{O}(z\tilde{M}D\log_2 D)$. Unlike Sec. 3.2.1, the accumulated cost of diagonalizing the tridiagonal matrix at each iteration using MR [81,83,84] only applies to the first recursion. This diagonalization cost is $\mathcal{O}(M^3) \sim 10^6$, which is very small compared to the cost of matrix-vector multiplications, e.g. $\mathcal{O}(z\tilde{M}D\log_2 D) \sim 10^{12}$ for $N = 24$.

### 3.4.2 Hybrid Lanczos-Chebyshev method for spectral functions

Here, we use the Chebyshev expansion of the resolvent operator of Eq. (19) to compute spectral functions directly in the frequency domain. This approach is inspired by a similar technique described in Ref. [49], where a kernel polynomial approximation based on Chebyshev polynomials is used. This approach was further exploited in Refs. [50,74], where Chebyshev expansions were combined with Matrix Product States (MPS) and DMRG to investigate one-dimensional strongly correlated systems. Yet, this technique has so far relied on the use of a kernel convolutions to damp Gibbs oscillations in the Chebyshev expansion. Here, we combine the numerically exact Chebyshev expansion of Eq. (19), which avoids the use of a kernel, with Lanczos. The key advantage of this approach is the rigorous control over resolution, a feature that is shared with the CPGF method that was described above in Sec. 3.2.3. In principle, the ideas of the method described below could be combined with MPS and DMRG as well, but such a task is outside the scope of this work.

Similarly to Lanczos, we start the procedure with the state $|\tilde{\phi}_0\rangle$ — obtained from two prior Lanczos recursions — and, instead of a third Lanczos recursion, we carry out a Chebyshev recursion to generate the polynomials $T_n(\tilde{h})$ using Eq. (18), while storing the moments

$$\mu_n = \langle\tilde{\phi}_0|T_n(\tilde{h})|\tilde{\phi}_0\rangle. \tag{47}$$

The autocorrelation response function is then obtained as follows:

$$\mathcal{A}(\omega,\eta) = -\langle \text{GS}|\hat{A}^\dagger \hat{A}|\text{GS}\rangle \sum_n \text{Im}[g_n(\tilde{z})]\mu_n\,, \tag{48}$$

with $\tilde{z} = (\omega + \varepsilon_m + i\eta - b)/a$, where $a, b$ are defined in Eq. (15) and we recall that $\varepsilon_m$ is the ground state energy density. This procedure has significant advantages. The first is that two moments can be obtained per matrix-vector multiplication, which implies that the the numerical effort is halved (other parameters being fixed). Thus, the computational effort is proportional to the number of iterations, $\tilde{N}_{\text{it}} = \tilde{N}_{\text{poly}}/2$, where $\tilde{N}_{\text{poly}}$ is the number of Chebyshev moments needed for convergence. This is derived by applying the product identity $2T_m(x)T_n(x) = T_{m+n}(x) + T_{m-n}(x)$ to the overlaps $\langle \tilde{\phi}_n|\tilde{\phi}_n\rangle$ and $\langle \tilde{\phi}_{n+1}|\tilde{\phi}_n\rangle$, where $|\tilde{\phi}_n\rangle = T_n(\tilde{h})|\tilde{\phi}_0\rangle$:

$$\begin{aligned}
\langle \tilde{\phi}_n|\tilde{\phi}_n\rangle &= \frac{1}{2}\langle \tilde{\phi}_0|\left(T_{2n}(\tilde{h}) + \mathbb{I}\right)|\tilde{\phi}_0\rangle \\
&= \frac{1}{2}(\mu_{2n} + \mu_0) \\
\langle \tilde{\phi}_{n+1}|\tilde{\phi}_n\rangle &= \frac{1}{2}\langle \tilde{\phi}_0|\left(T_{2n+1}(\tilde{h}) + T_1(\tilde{h})\right)|\tilde{\phi}_0\rangle \\
&= \frac{1}{2}(\mu_{2n+1} + \mu_1)\,.
\end{aligned} \tag{49}$$

For a given iteration, given a new $|\tilde{\phi}_n\rangle$, two moments can now be computed:

$$\begin{aligned}
\mu_{2n} &= 2\langle \tilde{\phi}_n|\tilde{\phi}_n\rangle - \mu_0\,, \\
\mu_{2n+1} &= 2\langle \tilde{\phi}_{n+1}|\tilde{\phi}_n\rangle - \mu_1\,.
\end{aligned} \tag{50}$$

Two remarks are now in order:

- $\tilde{N}_{\text{it}}$ is typically of the same order of magnitude as $\tilde{M}$, which guarantees that CPGF is at least as fast as Lanczos. In practice, we observe faster performance with CPGF. We attribute this to the possibility of better parallelization with CPGF because it requires half as many vector update loops. These loops are needed in order to carry out the Lanczos and CPGF recursions with only two vectors of dimension $D$ stored in memory. They incur a cost that, whilst not dominating over that of matrix-vector multiplications, still compares closely. In fact, the complexity of each of these loop is proportional to $D$. With Lanczos, the two steps of Eq. (4) involve two of these loops that need to be executed one after the other, in opposition to CPGF, which needs only a single loop vector update.

- CPGF can easily be modified to compute more general spectral functions (for which we might have $\hat{B} \neq \hat{A}^\dagger$) without a significant additional memory or computer time cost. The 3-vector memory cost is preserved because the vector used to generate $|\text{GS}\rangle$ during the second Lanczos recursion is not used in CPGF once the initial state, $|\tilde{\phi}_0\rangle$ is generated. Thus, this vector can be used to store $|\varphi\rangle \equiv \hat{B}^\dagger|\text{GS}\rangle/\sqrt{\langle \text{GS}|\hat{B}\hat{B}^\dagger|\text{GS}\rangle}$, which in turn can be used to compute the modified moments, $\mu'_n = \langle \varphi|T_n(\tilde{h})|\tilde{\phi}_0\rangle$ needed to Chebyshev-expand $\mathcal{C}^{\hat{B}\hat{A}}(\omega)$:

$$\mathcal{C}^{\hat{B}\hat{A}}(\omega,\eta) = -\sqrt{\langle \text{GS}|\hat{A}^\dagger \hat{A}|\text{GS}\rangle \langle \text{GS}|\hat{B}\hat{B}^\dagger|\text{GS}\rangle} \sum_n \text{Im}[g_n(\tilde{z})]\mu'_n\,. \tag{51}$$

In contrast, the continued fraction Lanczos approach does not work in the case $\hat{B} \neq \hat{A}^\dagger$. One must then resort to Eq. (40) to directly study the behavior of the time-domain correlator. This

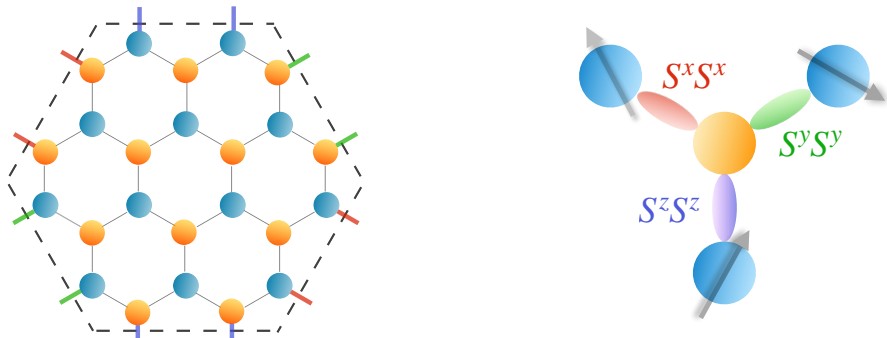

Figure 2: Left panel — 24-spin hexagonal cluster on the honeycomb lattice. The red, green and purple bonds illustrate the periodic boundary conditions. Right panel — Nearest neighbor bonds colored in red, green and purple (respectively $\gamma = x, y, z$ in our cartoon). The bond-directional character of the Kitaev interaction implies that to each bond corresponds a different type of interaction. Similarly to the case of the Ising model on the triangular lattice, where we have geometrical frustration, here we have exchange frustration due to the nature of the interaction and it is not possible to find a spin configuration that simultaneously minimizes the energy on all bonds.

leads to short time expansions with $M_t$ Lanczos vectors and the initial state $|\Psi(t=0)\rangle = |\tilde{\phi}_0\rangle$:

$$G^{\hat{B}\hat{A}}(\delta t) = \langle \mathrm{GS}|e^{iN\delta t\hat{h}}\hat{B}e^{-iN\delta t\hat{h}}\hat{A}|\mathrm{GS}\rangle \approx \sqrt{\langle \mathrm{GS}|\hat{A}^\dagger\hat{A}|\mathrm{GS}\rangle} \sum_{j=0}^{M_t} e^{-Ni(\varepsilon_j-\varepsilon_m)\delta t}\langle \mathrm{GS}|\hat{B}|\tilde{\psi}_j\rangle\langle\tilde{\psi}_j|\tilde{\phi}_0\rangle\,.$$

(52)

The eigenvectors of the tridiagonal matrix of Eq. (45), $\{\tilde{\mathbf{v}}_j\}$ give $\langle\tilde{\psi}_j|\tilde{\phi}_0\rangle = \tilde{v}_{j0}$. However, the overlaps of the type $\langle \mathrm{GS}|\hat{B}|\tilde{\psi}_j\rangle$ must be evaluated explicitly using the vector $\hat{B}|\mathrm{GS}\rangle$, which now has to be stored in memory separately, thus adding to the memory cost:

$$\langle \mathrm{GS}|\hat{B}|\tilde{\psi}_j\rangle = \sum_{i=0}^{M_t} \tilde{v}_{ji}\langle \mathrm{GS}|\hat{B}|\tilde{\phi}_i\rangle\,.$$

(53)

Moreover, we must update the initial state of the Lanczos expansion at each time interval, $|\Psi(t)\rangle$ using short time Lanczos expansions. Then, we re-compute the eigenvectors of a new tridiagonal matrix and re-evaluate $\langle \mathrm{GS}|\hat{B}|\tilde{\psi}_j\rangle$ for each time step. This process becomes computationally expensive very quickly since we may require a large number of time steps to capture important features of $G^{\hat{B}\hat{A}}(t)$. On the other hand, the CPGF treats the cases $\hat{B} \neq \hat{A}^\dagger$ and $\hat{B} = \hat{A}^\dagger$ on equal footing. Therefore, the CPGF is a general purpose approach, which accesses spectral functions for the case $\hat{B} \neq \hat{A}^\dagger$ using the same methodology and with the same computational complexity and memory requirements as the case $\hat{B} = \hat{A}^\dagger$.

## 4 Applications

In this section we apply the methods described in Sec. 3 to two generalized Kitaev models on the honeycomb lattice for a 24-spin hexagonal cluster with periodic boundary conditions (see left panel of Fig. 2): the K-H and the K-I models.

### 4.1 Kitaev-Heisenberg model

The K-H model combines Kitaev and Heisenberg interactions. The Hamiltonian can be cast as a sum over NN bonds $\langle i,j\rangle^\gamma$ on the honeycomb lattice (the superscript refers to the type of

bond, see right panel of Fig. 2):

$$\hat{H} = A \sum_{\langle i,j \rangle_\gamma} \left( K \hat{S}_i^\gamma \hat{S}_j^\gamma + J \hat{\mathbf{S}}_i \cdot \hat{\mathbf{S}}_j \right). \tag{54}$$

with $K = \sin \varphi$, $J = \cos \varphi$, and where $\gamma = x, y, z$ is one of the three types of bond on the honeycomb lattice. We use the conventions of Ref. [5]: $\varphi \in [0, 2\pi]$ parameterizes the strength of each term and ensures that all possible ratios of the Heisenberg and Kitaev interactions are considered, and an overall energy scale is defined and set to unity throughout: $A = \sqrt{J^2 + K^2} \equiv 1$. The Kitaev interaction is bond-directional, i.e. for each distinct type of bond $\gamma = x, y, z$, there is a correspondent interaction (respectively $S^x S^x$, $S^y S^y$, $S^z S^z$), as shown on the right panel of Fig. 2. The calculations presented throughout the next subsections serve two purposes: to prove that the classes of methods considered in this work are consistent and correctly capture the physics of frustrated quantum magnets and, more importantly, to show that the Chebyshev polynomial-based approaches offer significant advantages in terms of performance, stability and generality, especially when combined with Lanczos algorithms.

### 4.1.1 Zero-temperature consistency and microcanonical approaches

Firstly, we reproduced the results of Refs. [3–5] using ED (via the "ground state" Lanczos algorithm), as used in the original references. Then, we recovered these results using both microcanonical and canonical approaches based on Lanczos, TPQ and Chebyshev recursions.

In order to bench-mark the microcanonical approaches — MCLM, MTPQ and CPGF — we set the ground state energy obtained from Lanczos as the target energy. Crucially, MTPQ and CPGF (and their canonical counterparts) require an estimate of the maximum eigenvalue as well. Although we could have applied Lanczos to the operator $-\hat{h}$ to obtain it, the Hamiltonian of Eq.(54) happens to have a symmetry that can be used to avoid this additional computation: $-\hat{h}(\varphi) = \hat{h}((\varphi + \pi) \bmod (2\pi))$. Thus, the maximum eigenvalues can be obtained by simply reorganizing the minimum eigenvalues as a function of $\varphi$ and switching their sign (see Fig. 3). These are the minimum and maximum energies that are then used as inputs to the TPQ and Chebyshev methods. We remark that even though the ground state energy can be accurately estimated solely using MTPQ, the method requires an upper bound on the maximum eigenvalue (that one would normally obtain from Lanczos anyway). Moreover, MTPQ has much slower convergence to the ground state than Lanczos, as we will also show later, so it is simply more efficient to use Lanczos to obtain extremal eigenvalues. Still, MTPQ provides us with a useful consistency check, while being less memory-intensive — requiring 2 rather than 3 vectors of size $D$ — and giving access to good approximations of finite temperature states.

The results we present throughout this section are for the ground state energy density and NN spin–spin correlation of the K-H model. The NN spin–spin correlation is used for our benchmark for two reasons. Firstly, in Ref. [5], the authors show that longer-range correlations vanish in the vicinity of the spin liquid phases. Given that we are particularly interested in this region of the phase diagram, it is reasonable to focus on NN correlations. Secondly, the step-like behavior of the NN spin–spin correlation coincides with quantum critical points [5]. Moreover, the behavior of the NN correlation as a function of the temperature is intimately connected to peaks in the specific heat [10] — that we also compute using Lanczos, TPQ and Chebyshev — and that are particularly relevant experimentally [89].

For each value of $\varphi$, we computed the minimum eigenvalue, or ground-state energy of the Hamiltonian, $\varepsilon_{\mathrm{GS}}$, using the Lanczos ED technique. In the top panel of Fig. 4, we show these results, along with the second derivative $-d^2 \varepsilon_{\mathrm{GS}}/d\varphi^2$, which accurately detects quantum phase transitions between magnetically ordered phases (Néel, stripy and zigzag antiferromagnets and a ferromagnet) and two spin liquid phases, which we refer to as "antiferromagnetic"

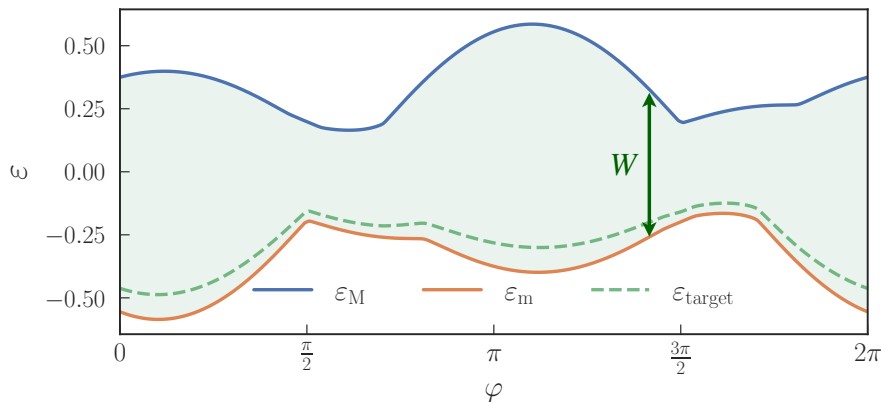

Figure 3: Minimum ($\varepsilon_m$) and maximum ($\varepsilon_M$) eigenvalues of the K-H Hamiltonian on a 24-spin hexagonal cluster on the honeycomb lattice, computed with the ground state variant of Lanczos ED. The energy is normalized to the number of spins and the shaded green area represents the spectrum width $W = \varepsilon_M - \varepsilon_m$ for each value of $\varphi$. $\varepsilon_{\text{target}} = \varepsilon_m + 0.1W$ are target energies used later to compare MCLM and CPGF.

($\varphi \sim \pi/2$) and "ferromagnetic" ($\varphi \sim 3\pi/2$) QSL phases, or AFQSL and FQSL in short. The phase boundaries we obtained are summarized in Table 1 and agree well with Ref. [5]. The Lanczos algorithm shows small statistical fluctuations, enabling one to use a single realization of the initial random state across the entire phase diagram. As mentioned in Sec. 3.2.2, a single random vector suffices to achieve good accuracy in the MTPQ approach as well due to the self-averaging properties of the TPQ estimator. In fact, error bars obtained with more realizations are negligibly small and thus are not shown in our plots.

We now move gears to the spin correlator. Figure 4 shows excellent agreement between the spin–spin correlation computed with Lanczos and MTPQ. As mentioned earlier, the step discontinuities in the correlator signal the quantum phase transitions of the K-H model [3–5]. MTPQ achieves its maximum effective resolution at low-temperatures (i.e. large $N_{\text{it.}}$) where Fig. 4 shows that it accurately approximates the ground state [34–36,90]. These methods are designed to reconstruct the ground state in a recursive fashion. Thus, it is perhaps not too surprising that the energy and N-N spin correlation can be both computed with great accuracy provided enough iterations are completed. Also shown in the bottom panel of Fig. 4 is the behavior of the spin correlation for the target energies larger than the ground state by 10% of the spectrum width, obtained with CPGF (the MCLM results coincide, and thus are omitted to avoid unnecessary clutter on the figure). These excited states are still close enough to the ground state that the spin correlation preserves its general shape, albeit with a considerable broadening of its sharper features.

Table 1: Phase boundaries of the K-H model on a 24-spin hexagonal cluster obtained with Lanczos.

| QCP | $\varphi/\pi$ | QCP | $\varphi/\pi$ |
|---|---|---|---|
| Néel-AFQSL | 0.4940 | Ferromagnet-FQSL | 1.4500 |
| AFQSL-Zigzag | 0.5064 | FQSL-Stripy | 1.5364 |
| Zigzag-Ferromagnet | 0.8156 | Stripy-Néel | 1.7048 |

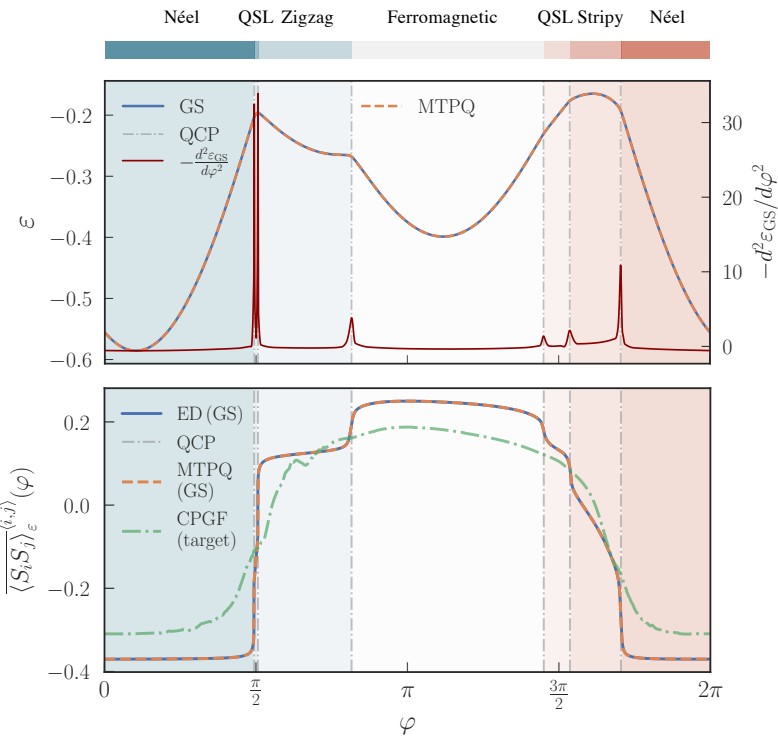

Figure 4: Top panel — Ground state energy density obtained with Lanczos ED (solid blue line) and MTPQ (dashed orange line). The solid red line is minus the second derivative of the Lanczos ED curve obtained via finite differences. Its peaks signal the quantum critical points labeled "QCP". Bottom panel — Ground state nearest neighbor spin–spin correlation computed with Lanczos ED compared with MTPQ. The correlator is also computed with CPGF for states with target energy $\varepsilon_{\text{target}}$. The symbol $\overline{\phantom{xx}}^{\langle i,j \rangle}$ denotes an average over all nearest neighbors $\langle i,j \rangle$.

The top panel in Fig. 5 summarizes a careful convergence study aimed at understanding how the optimal number of iterations, $N_{\text{it}}^*$,[5] depends on the microscopic details of the model. Our simulations show that convergence is relatively fast in the purely Heisenberg limits ($\varphi = 0, \pi$) and becomes slower as we approach the quantum phase transitions and, in particular the Kitaev limits ($\varphi = \pi/2, 3\pi/2$), in both Lanczos and MTPQ. Note that in spite of the strong dependence of $N_{\text{it}}^*$ with $\varphi$, Lanczos converges considerably faster throughout the K-H parameter space, requiring about 2 orders of magnitude less iterations for convergence than MTPQ.

Next, we address convergence around the quantum critical points near the Kitaev limits in more detail; see Fig. 5 (bottom panel). The convergence of the Lanczos and TPQ estimators for the spin–spin correlation is found to slow down notoriously as the Kitaev term on the Hamiltonian becomes dominant ($|K|/|J| \gg 1$) and the ground state energy per site $\varepsilon_{\text{GS}}(\varphi)$ and the NN spin–spin correlation $\overline{\langle S_i S_j \rangle}_{\text{GS}}^{\langle i,j \rangle}(\varphi)$ start to differ significantly. As a result of the slower convergence away from the purely Heisenberg points, the computational effort grows substantially, notably so close to the Kitaev points. For example, for $\varphi \simeq 0.506\pi \iff |K|/|J| \simeq 53$, MTPQ requires around $4 \times 10^4$ iterations for optimal convergence as defined earlier to be achieved.

Figure 6 shows the energy density and spin correlation function of the 24-site cluster across the phase diagram calculated with CPGF. We compare the latter with MCLM, which also has

---

[5]Here, optimal convergence is defined as a variation of less than $10^{-9}$ in the energy density between consecutive iterations.

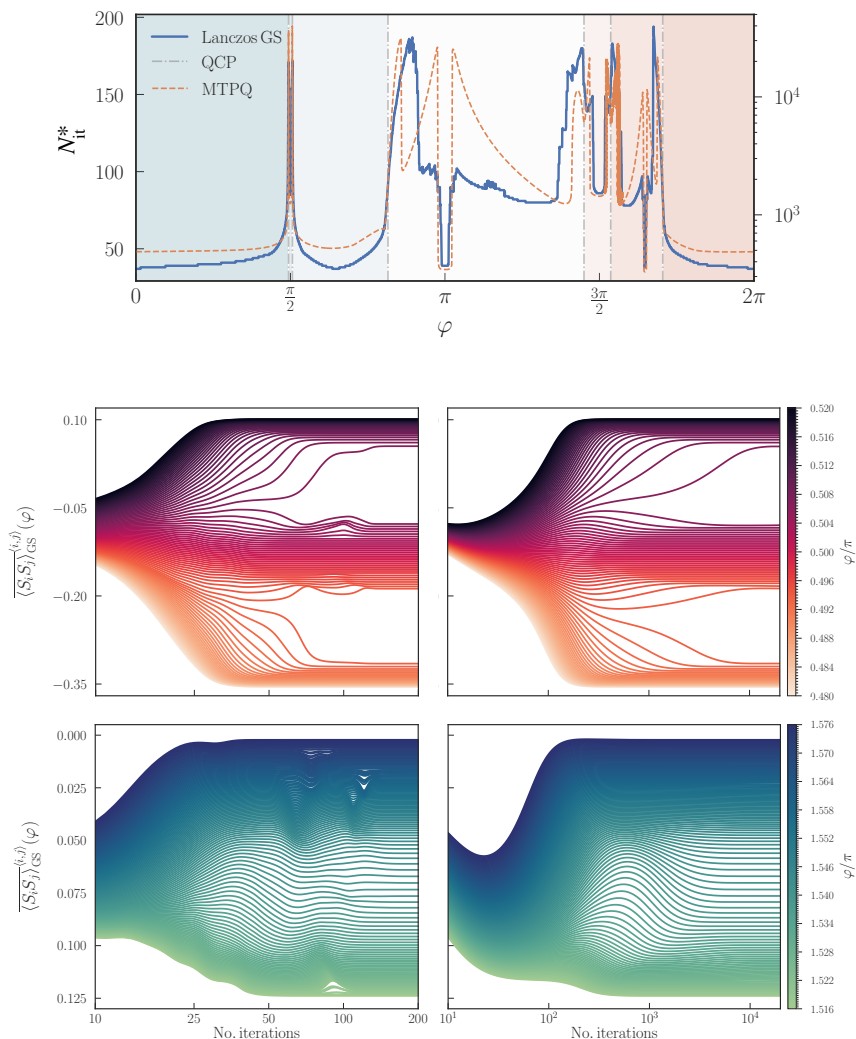

Figure 5: Convergence in computations of ground state N-N spin–spin correlations. Top panel – Dependence of $N_{it}^*$ with $\varphi$ in Lanczos and MTPQ -(left and right vertical axes, respectively). Bottom panel – Convergence of Lanczos (left) and MTPQ (right) around the Kitaev limits: AFQSL on top and FQSL on the bottom .

the ability to probe an input target energy. We also establish the accuracy of the STE of Eq. (22) using CPGF. A detailed analysis shown in Appendix A confirms that the standard deviation of the estimate for the N-N spin–spin correlation function scales as expected, i.e. as the inverse square root of the number of realizations of the initial random state.

These results also show that finer CPGF resolutions are needed to probe the phase diagram around the Kitaev points ($\varphi = \pi/2, 3\pi/2$). On the contrary, near the Heisenberg limits ($\varphi = 0, \pi$), convergence occurs with comparatively coarse resolution. A useful feature of the CPGF method is that the optimal number of Chebyshev iterations, $N_{poly}^* = N_{poly}^*(\eta)$, follows a predictable pattern: it is roughly proportional to the spectrum width and inversely proportional to the required resolution [51, 55]. The results reported in Appendix B confirm this expected behavior. We find that the convergence speed of MCLM and CPGF compares very differently throughout the phase diagram. MCLM seems to show unpredictable behavior as some points of the phase diagram require significantly more computational effort than others. CPGF behaves more intuitively, requiring a greater effort for points close to the phase transi-

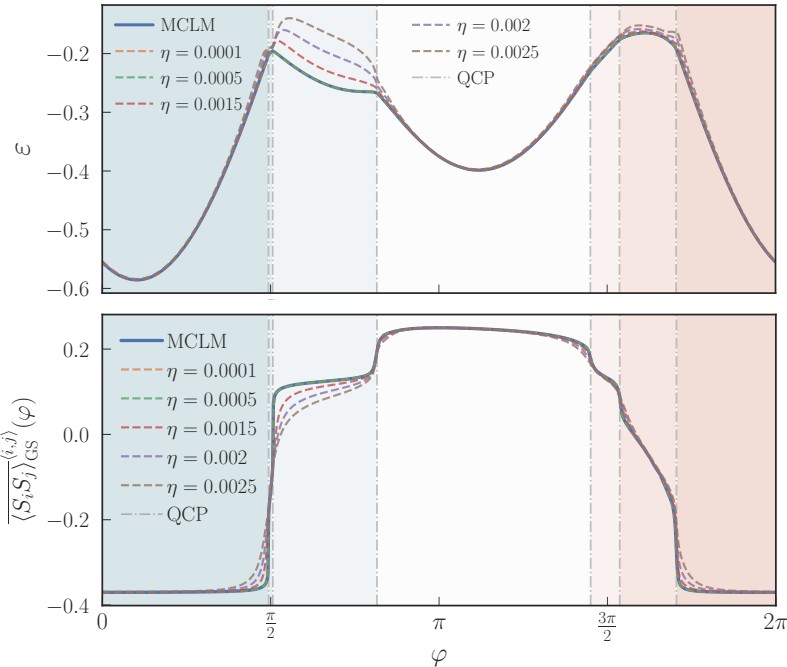

Figure 6: Comparative study between CPGF and MCLM. Top panel — Ground state energy density obtained with MCLM (solid blue line) and CPGF (dashed lines) with varying resolution as a function of $\varphi$. The two methods show excellent agreement provided that the CPGF resolution is sufficiently fine. Bottom panel — Ground state N-N spin–spin correlation as a function of $\varphi$. We used a single realization of the initial random state for both methods. The results shown here also match the Lanczos and MTPQ results shown in Fig. 4.

tions, which need finer resolutions and, consequently more iterations and thus more computer time. Although the CPGF approach is significantly faster in some regions of the K-H parameter space, the MCLM approach is faster close to the transitions (see Appendix B). Nevertheless, when targeting the ground state, CPGF is 25% faster on average. For the excited states, the difference in performance is much more pronounced and the results of Appendix B show that CPGF is the method of choice due to its faster overall convergence (about an order of magnitude less CPU time required).

Next, we bench-mark the MCLM and CPGF methods around the Kitaev limits in detail, i.e. for spreads of $\varphi$-values in the intervals $[0.4800, 0.5200]\pi$ and $[1.5160, 1.5760]\pi$. To assess convergence in the K-H model, we carefully track the evolution of the correlation function as more iterations are completed with each of the two methods.

In Fig. 7, we show the dependence of the NN spin correlation on the number of iterations in the MCLM and CPGF approaches. Here, we focused on a range of resolutions in the interval $[2.5 \times 10^{-5}, 5 \times 10^{-4}]$, but only plotted the curves corresponding to the energy resolutions that yield convergence. In this case, we consider that a given resolution yields convergence when the energy density and spin correlation no longer change appreciably as finer resolutions are considered. To ensure that convergence has indeed occurred, we compute the difference between the energy density for two consecutive resolutions and ensure that the difference is smaller than the resolution itself. As it turns out, convergence is achieved for $N_{\text{poly}} \sim 10^2 - 10^3$ depending on the value of $\varphi$. Figure 7 confirms that enhanced resolutions are crucial in the vicinity of quantum critical points. Moreover, it also shows that MCLM and CPGF have complementary convergence behavior. The top left panel shows that MCLM converges quicker around

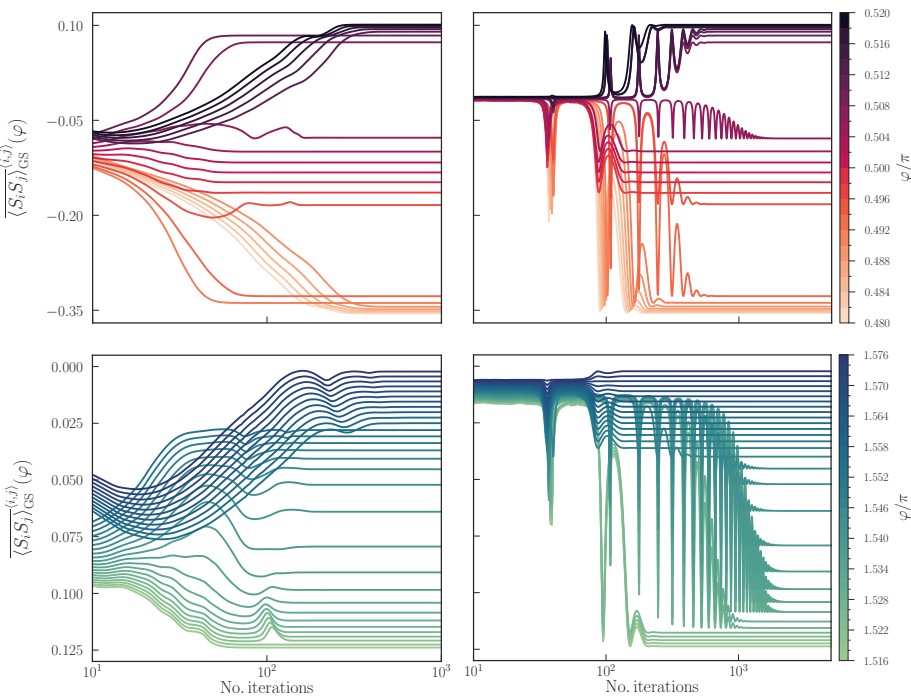

Figure 7: Convergence of the ground state nearest neighbor spin–spin correlation computed with MCLM (left) and CPGF (right) around the Kitaev limits.

$0.500\pi$, but dramatically slows down around $0.480\pi$ and $0.520\pi$. The top right panel shows that CPGF has more consistent and faster convergence, with only the points closest to the transition ($0.494\pi$, $0.506\pi$) requiring more iterations due to the need for a finer resolution. The bottom panels exhibit a similar tendency. The points near $1.576\pi$ display faster convergence with CPGF than with MCLM, while the curves in the center of the two plots show that convergence is faster with MCLM due to the increased resolution that is needed with CPGF. Here, we emphasize that we found each iteration to be faster with CPGF due to a lower number of necessary operations than with MCLM. Thus, CPGF is still faster even in situations where the number of iterations required for convergence is similar, e.g. for the points near $1.516\pi$ in the bottom panels. The main contribution to the extra time per iteration in MCLM is the action of $\hat{h}^2$ upon a state at each iteration. Since the Hamiltonian is generated "on-the-fly", one must act with $\hat{h}$ twice in order to obtain the action of $\hat{h}^2$. Thus, each iteration incurs an extra cost due to the additional matrix-vector multiplication, which is the most computationally expensive operation in these methods.

### 4.1.2 Canonical approaches: Finite temperature

In what follows, we discuss the performance of the finite-temperature approaches introduced in Sec. 3.3. We also include the microcanonical variant of the TPQ method in this analysis and confirm that the temperature corresponding to each energy density can be accurately estimated as explained in Ref. [34]. For concreteness, we focus on the Kitaev point at $\varphi = 1.5\pi$ and restrict the temperature to the range $T = [0.004, 24]$ in units of the Kitaev coupling, which suffices to capture the relevant features of the model.

We set out to obtain the NN spin correlation, specific heat and entropy with all methods, with a careful convergence analysis. Thus, we compute the minimum (i.e., optimal) number of iterations such that the target functions are reliably captured within the desired accuracy at all

Table 2: Number of iterations (or Chebyshev polynomials) required for convergence for each method and CPU time per core per iteration.

| Method | No. iterations | CPU time per core per iteration / s |
|---|---|---|
| MTPQ ($e_{\text{upper}} = e_M$) | 1200 | 0.56 |
| MTPQ ($e_{\text{upper}} = 35e_M$) | 22000 | 0.57 |
| FTLM | 200 | 2.05 |
| CTPQ | 500 | 0.58 |
| FTCP ($5 < N_{\text{poly}} < 20$) | 7515 | 0.98 |

temperatures. Naturally, the relevant convergence parameters need to be adjusted separately for each method due to their different characteristics. These are summarized in Table 2. It is important to note that the results presented below not only agree with each other, but also with QMC and exponential tensor renormalization group (XTRG) studies of Ref. [10], thus supporting the validity of our implementation for all methods. The specific heat is computed as follows: $c = N\beta^2(\langle \hat{h}^2 \rangle - \langle \hat{h} \rangle^2)$. Similarly to Ref. [91], the entropy density is computed by integrating $c/T$. We perform the integral numerically using Simpson's rule.

At first sight, Table 2 seems to indicate that FTLM is the most efficient method. However, notice that in Fig. 8, FTLM shows large low-temperature fluctuations, reaching about 26 times the fluctuations of FTCP (calculated from root-mean-square deviations). The inset of Fig. 8 shows that these fluctuations are larger for FTLM than for the other methods throughout the temperature range. Here, we note that the same initial random vectors are used in all methods so that statistical fluctuations can be directly compared. Moreover, the total computational cost is dictated not only by the number of iterations, but also by the number of matrix-vector multiplications per iteration. This number is higher for FTLM than FTCP, leading to about twice the average computer time per iteration (2.05 seconds with FTLM compared to 0.98 seconds with FTCP).

Figure 9 shows a very important shortcoming of the CTPQ method. Unlike the NN spin correlation, the specific heat has an important feature that cannot be reproduced with CTPQ,

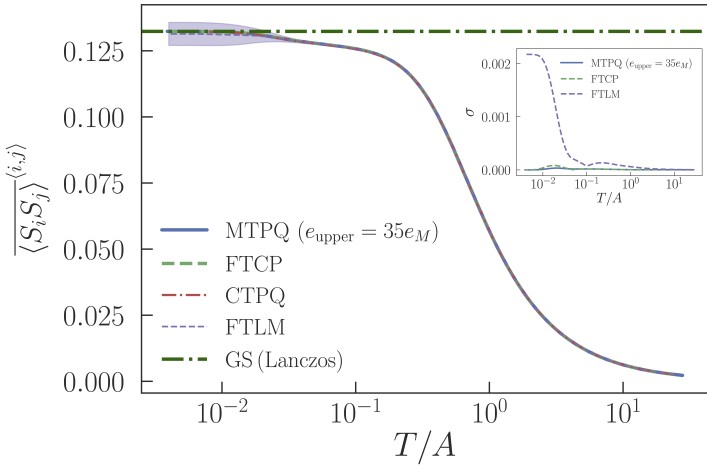

Figure 8: Finite temperature N-N spin correlator computed with MTPQ, FTCP, CTPQ and FTLM for the 24-site cluster at the Kitaev limit $\varphi = 3\pi/2$. We used 50 random vector realizations in all cases. Error bars are negligibly small, except for FTLM. Inset: Standard deviation of the N-N spin correlator. CTPQ is not shown because the fluctuations are comparable to MTPQ.

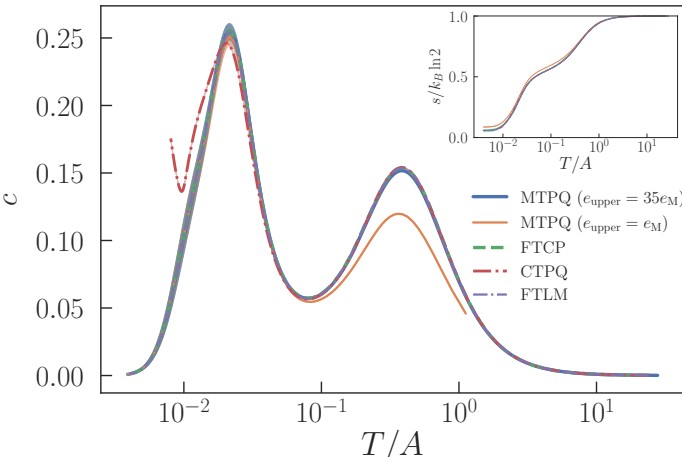

Figure 9: Specific heat (entropy density on the inset) computed with MTPQ, FTCP, CTPQ and FTLM. We use 50 realizations of the initial random state for all methods. The error bars are negligibly small in most of the temperature range, except for temperatures below that of the low-temperature peak. The error bars are of comparable size for every method (except for CTPQ, for which their meaning is ill-defined due to a numerical instability). Model parameters as in Fig. 8.

namely the decay to zero of the low-temperature peak as $T \to 0$. The limitation is related to the need to consider more terms in the summation of Eq. (31) so as to achieve convergence. We find that when we considered 500 terms, we reached the limit of machine precision (80-bit floating-point on an Intel Core i5 processor) before reaching enough accuracy. The CTPQ results gradually lose accuracy as temperature decreases and at some point between $10^{-1}$ and $10^{-2}$, they are no longer reliable. The entropy also shows signs of the numerical instability of CTPQ (see inset of Fig. 9). FTCP avoids this instability via the numerical stabilization procedure described in Sec. 3.3.3. The operator break-up into the product of Eq. (34) ensures that each Chebyshev expansion in Eq. (36) is stable. The arguments of the fast growing modified Bessel functions are guaranteed to remain in check because they can be controlled via the inverse temperature step. This is to be contrasted with Eq. (31), where the also fast growing functions of the inverse temperature and the number of iterations are not controlled, leading to the numerical instability visible in Fig. 9.

Having discussed the limitations of FTLM (large low-temperature fluctuations) and CTPQ (numerical instability), we now move on to MTPQ. An often ommited detail in the literature is that the upper bound on the maximum eigenvalue of the Hamiltonian density, $e_M$, given as an input to MTPQ determines: i) the temperature range that is covered, and ii) how much accuracy is achieved for a given temperature. The strong influence of $e_M$ can be traced back to the evolution of the TPQ energy density distribution with the number of iterations [34]. We found that in order to cover the desired temperature range with enough accuracy in MTPQ, we had to increase the upper bound to 35 times the maximum eigenvalue of the Hamiltonian density (computed with Lanczos). We consider that enough accuracy has been reached when the specific heat and entropy computed with MTPQ match the FTCP and FTLM results.

The results of Fig. 9 indicate that the disparity between MTPQ with $e_{\text{upper}} = e_M$ and the other methods is solely due to insufficient accuracy at high temperature. While MTPQ captures the high-temperature peak of the specific heat even with $e_{\text{upper}} = e_M$, the amplitude of this peak is severely underestimated. As more iterations are completed, the accuracy of the method increases and the low-temperature peak matches the results of other methods better, but still not perfectly. Moreover, even though the MTPQ result with $e_{\text{upper}} = e_M$ captures the general

behavior of the entropy density, there is not enough accuracy because of accumulated error from the lack of accuracy at high temperature and the reduced temperature range (note that we compute the entropy as an integral over temperature). The other methods match perfectly and show comparable fluctuations. Since there are only small error bars of approximately the same size for all methods at low temperature, the disparity can only be due to insufficient accuracy for MTPQ with $e_{\text{upper}} = e_M$. By probing higher values of $e_{\text{upper}}$, we find that is necessary to input $e_{\text{upper}} \approx 35 e_M$ for the high and low-temperature peaks to reproduce the correct behavior. When we consider this upper bound, MTPQ requires about twice the total computer time compared to FTCP in order to achieve comparable results (14268 versus 7376 seconds per core). Even though the number of iterations is nearly 3 times higher in MTPQ compared to FTCP, the former requires 40 % less matrix-vector operations per iteration. Still, FTCP remains about two times faster due to its significantly faster convergence.

Finally, notice that for both the specific heat and the entropy in Fig. 9, FTLM does not show the low-temperature fluctuations of Fig. 8. We attribute this to the fact that Lanczos methods are designed to quickly achieve an approximation of the Hamiltonian in a subspace restricted to the ground state and low-lying excitations. The specific heat and the entropy are calculated solely in terms of averages of the Hamiltonian density, $\langle \hat{h} \rangle$ and $\langle \hat{h}^2 \rangle$. Thus, convergence is better for these quantities than for more general observables, such as the NN spin correlation, which are not as closely associated with the Hamiltonian.

## 4.2 Kitaev-Ising model

The K-I model combines Kitaev and Ising interactions. The model Hamiltonian is:

$$\hat{H} = -\sum_{\langle i,j \rangle^\gamma} \left( K_\gamma \hat{\sigma}_i^\gamma \hat{\sigma}_j^\gamma + J_I \hat{\sigma}_i^z \hat{\sigma}_j^z \right), \tag{55}$$

where we use the parametrization of Ref. [61]. We allow not only for the isotropic case, but also a particular type of anisotropy, for which we have: $K_x = 1 - 2\alpha/3$, $K_y = K_z \equiv K_{yz} = \alpha/3$, where $\alpha \in [0, 1.5]$ is a parameter and the energy scale is set by $K_x + K_y + K_z = 1$, $K_\gamma \geq 0$.

In Ref. [61], the authors study this model for the 24-spin hexagonal cluster of the left panel of Fig. 2 using Lanczos at $T = 0$. We start by reproducing some of their results so as to validate our implementation. Then, we present a new study displaying dynamical signatures of the phase transitions described in Ref. [61]. These signatures are found in the $\sigma^z$ spin susceptiblity, which we obtain using both the Lanczos and the hybrid Lanczos-Chebyshev (HLC) approach that we introduced in Sec. 3.4.2.

### 4.2.1 Lanczos bench-mark

Figure 10 summarizes our Lanczos results and reproduces those of Ref. [61]. The top panel shows two steps in the first derivative of the ground state energy density for fixed $\alpha = 0.7$ and varying $J_I \in [10^{-4}, 10]$. Correspondingly, the center panel displays two peaks in the second derivative. These steps/peaks indicate two quantum phase transitions. From left to right, the first quantum critical point separates a quantum spin liquid (blue) from a nematic (green) phase and the second one separates the latter from a ferromagnetic phase (red). Similarly to how the steps in the spin-spin correlation signal transitions for the K-H model (see bottom panel of Fig. 4), the mean squared magnetization also signals transitions in the K-I model, with its value approaching saturation very quickly as we enter the ferromagnetic phase.

### 4.2.2 Finite temperature: Comparing thermal pure quantum states and Chebyshev

In section 4.1.2, we bench-marked the FTCP approach by computing the temperature dependence of the specific heat, entropy density and NN spin correlation. In principle, MTPQ could

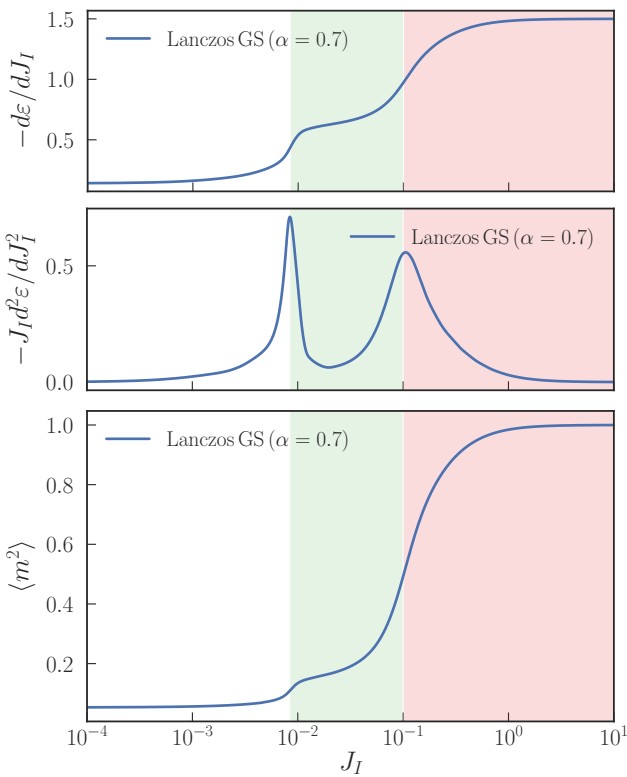

Figure 10: Top and center panels – Negative first (top) and second (center) derivatives of the Lanczos ground state energy density. Bottom panel – Mean square of the magnetization, $\langle m^2 \rangle \equiv \langle [(1/N)\sum_i \sigma_i^z]^2 \rangle$. These results match those of Ref. [61] and illustrate the transitions between ferromagnetic (red), nematic (green) and quantum spin liquid (blue) phases found in Ref. [61]. Here, a single realization of the initial random state was found to be sufficient due to negligible statistical fluctuations.

be the most viable competitor of FTCP (despite being a microcanonical method) because it avoids the shortcomings of FTLM and CTPQ. Yet, we found that MTPQ required about twice the computer time of FTCP in the context of the K-H model. Here, we further bench-mark FTCP by considering the K-I model. We also seek to clarify whether FTCP outperforms MTPQ in terms of computer time for a different model, which would suggest that the advantages of FTCP are not problem-dependent. We start by recovering the MTPQ results of Ref. [61] using our own implementation. Then, we repeat the calculation using FTCP. These results — shown in Fig. 11 — are for two specific points of the phase diagram: $(\alpha, J_I) = \{(0.7, 0.001), (0.7, 0.03)\}$. Going back to Fig. 10, we can see that these two points are located within the Kitaev quantum spin liquid and nematic regions of the phase diagram, respectively.

In Ref. [61], the authors remark that the well known results for the pure Kitaev model that we recovered in Figs. 8 and 9 remain qualitatively valid even when $J_I \neq 0$, and even within the nematic phase (see right panels of Fig.11 with results for $(\alpha, J_I) = (0.7, 0.03)$). The specific heat has a two-peak structure, corresponding to a two-step release of entropy. At high temperature ($T \sim 0.5$), the Majorana fermions c (defined in [61]) release their entropy ($0.5k_B \ln 2$). The other half of the entropy is released by $Z_2$ fluxes at low temperature ($T \lesssim 10^{-2}$) [8]. The high temperature crossover coincides with an enhancement of the expectation of the kinetic

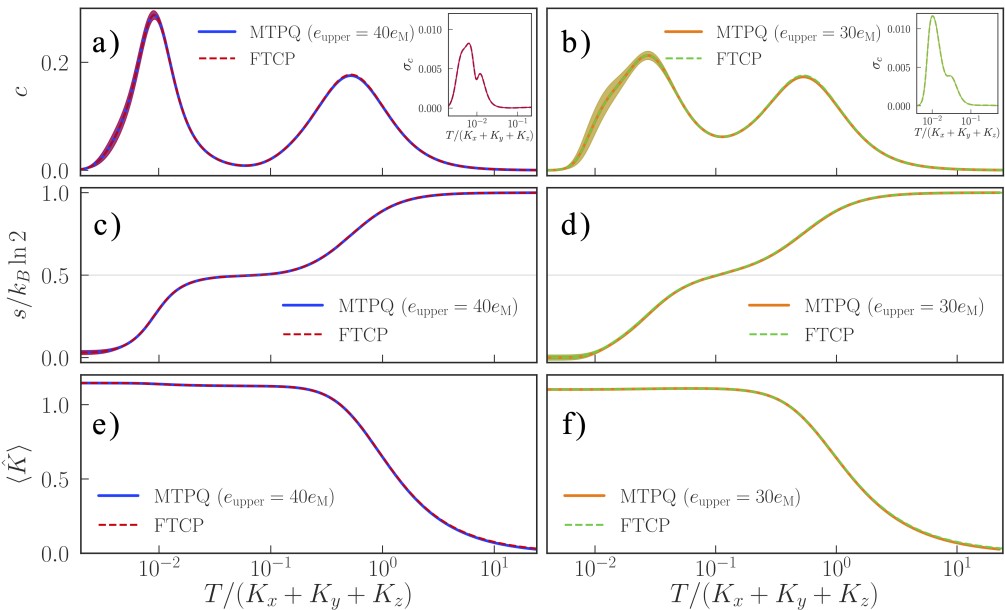

Figure 11: MTPQ and FTCP results for the K-I model on a 24-spin cluster with $\alpha = 0.7$ and $J_I = 0.001$ (left panels) and $J_I = 0.03$ (right panels). We used 50 and 100 random vector realizations for $J_I = 0.001$ and $J_I = 0.03$, respectively. Panels a) and b) show the specific heat, with a zoom-in on the standard deviations — from which the error bars are derived — at low temperature; panels c) and d) show the entropy density, with the grey lines marking $s = 0.5k_B \ln 2$; panels e),f) show the finite temperature expectation of the kinetic energy of the Majorana fermions $c$.

energy of the Majorana fermions $c$, defined in terms of the spin operators as follows:

$$\hat{K} = \frac{2}{N} \sum_{\gamma=x,y} \sum_{\langle j,k \rangle_\gamma} \hat{\sigma}_j^\gamma \hat{\sigma}_k^\gamma . \tag{56}$$

The features mentioned above are all apparent in Fig. 11 and our MTPQ and FTCP results match nearly perfectly. The striking difference between the two approaches is that once again, FTCP cuts the computer time in approximately half. To be more precise, for $J_I = 0.03$, FTCP is around 2.1 times faster and for $J_I = 0.001$, it is around 1.8 times faster. Table 3 summarizes these differences in computer time.

Figures 11 a),b) confirm the two-peak behavior of the specific heat. In the right panel ($J_I = 0.03$), statistical fluctuations are more apparent and even doubling the number of random vectors compared with the case $J_I = 0.001$ (going from 50 to 100), we find that statistical fluctuations remain higher. This is illustrated in the low temperature behavior of the standard deviation of the specific heat estimator, which is shown on the insets. MTPQ and FTCP show identical statistical properties, with these standard deviations matching remarkably well. Notice that the optimal value for the upper bounds on the maximum eigenvalue of Hamiltonian used in MTPQ were different in each case ($40e_M$ for $J_I = 0.001$ and $30e_M$ for $J_I = 0.03$). These were chosen so as to ensure enough accuracy throughout the chosen temperature ranges ($T \in [0.002, 24]$ for $J_I = 0.001$ and $T \in [0.004, 24]$ for $J_I = 0.03$, in units of $K_x + K_y + K_z$). Figures 11 c),d) show the two-step release of entropy. Compared with

Table 3: Average CPU times, $t_{\text{CPU}}$, for MTPQ and FTCP calculations for the K-I model on a 24-spin hexagonal cluster with $\alpha = 0.7$ and $J_I = 0.001, 0.03$.

| $J_I$ | $t_{\text{CPU}}$/ hours | |
|---|---|---|
| | MTPQ | FTCP |
| 0.001 | 142 | 77 |
| 0.03 | 61 | 29 |

the pure Kitaev case of Fig. 9, the left panel (c) shows a much more pronounced plateau-like behavior between $T = 10^{-1}$ and $T = 10^{-2}$, ending in an abrupt decrease of entropy. In contrast, the right panel (d) shows no plateau at all, with a gentler decrease in entropy between $T = 10^{-1}$ and $T = 10^{-2}$. This is a manifestation of the intrinsic differences between the two liquid phases (Kitaev QSL and nematic). Finally, figures 11 e),f) illustrate the high temperature enhancement of the kinetic energy of the Majorana fermions $c$, a behavior that is shared between the two phases. Here, statistical fluctuations are very small for both MTPQ and FTCP, with negligible error bars.

### 4.2.3 Dynamics: Hybrid Lanczos-Chebyshev approach

Lastly, we present novel results that elaborate on the picture of the K-I system that was outlined in Ref. [61]. We find that the signatures of the quantum phase transitions are present not only in static quantities, such as the energy and squared magnetization, but also in the dynamical spin susceptibility. This spectral function is obtained by considering the relevant observable in Eq. (48) to be the Fourier-transformed spin operator, i.e. $\hat{A} = \sum_{\mathbf{r}} e^{-i\mathbf{q}\cdot\mathbf{r}} \hat{S}_{\mathbf{r}}^z / \sqrt{N}$, where $\mathbf{r}$ is a position on the lattice and $\mathbf{q}$ is the wave vector.

In Fig. 12, we show the variation of the $\mathbf{q} = (0,0)$ dynamical spin susceptibility, $S_\Gamma(\omega)$, with the model parameter $J_I$. These results are obtained with the hybrid Lanczos-Chebyshev

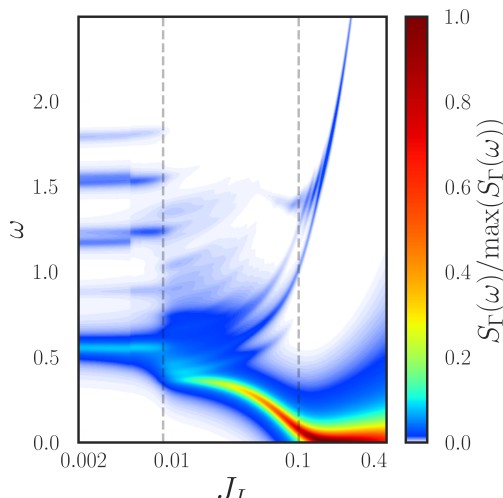

Figure 12: Dynamical spin susceptibility of the K-I system for varying $J_I$ and $\alpha = 0.7$, normalized to its maximum value. The phase transitions are marked as gray dashed lines. The results obtained with the Lanczos and HLC methods are identical. The white space corresponds to a vanishing $S_\Gamma(\omega)$, as shown on the bottom of the color bar.

(HLC) method. The initial Lanczos run is stopped when the variation between the ground state energy density computed for consecutive iterations is less than $10^{-9}$ in units of $K_x + K_y + K_z$. We compute 3000 Chebyshev moments, which is enough to achieve convergence for all the values of $\eta$ considered. Specifically, for each $J_I$, the resolution parameter is fixed to 0.1% of the spectrum width, which translates to values in the interval $\eta \in [0.0011, 0.0036](K_x + K_y + K_z)$. In terms of statistical sampling, we find that lower values of $J_I$ require more random vectors for the peaks of the dynamic susceptibility to be resolved satisfactorily. Thus, for $J_I < 0.0057$, we use 16, rather than the 4 initial random vectors that we use for $J_I > 0.0057$. Our interpretation of these results is in line with a similar reasoning for the K-H model presented in Ref. [5], albeit with a crucial difference due to the specifics of the liquid-to-liquid transition. In the ferromagnetic limit ($J_I \gtrsim 10^{-1}$), the $\omega = 0$ component dominates because of the strong ferromagnetic correlations. As $J_I$ is lowered and the transition to the nematic phase occurs, the gapless magnon mode gradually turns into a gapped mode. Moreover, there is a proliferation of sharp well-defined excitations in the nematic phase, which abruptly collapse onto a smaller set of modes as the transition to the Kitaev phase occurs ($J_I \sim 10^{-2}$). This rapid change in the spin susceptibility is consistent with the $T = 0$ first order topological phase transition described in Ref. [61]. The gap is found to peak for the Kitaev liquid, at which point the lowest-$\omega$ mode occurs for a larger $\omega$ than in the nematic phase.

We close this section with a note on computational resources and memory cost. Throughout the paper, our calculations are done using Intel Xeon Gold 6138 processors running at 2 GHz and each simulation requires between 0.5 and 1 GB of memory. All Chebyshev methods showed improved performance. In the case of the dynamics studies of this section, we found that under the exact same circumstances, our HLC method is 33 % faster than the continued fraction Lanczos approach. This estimate was obtained as follows. We reproduced a previously known result for the dynamical spin susceptibility from Ref. [5] using the same number of Lanczos iterations/polynomials and averaging over $\sim 400$ simulations, each running with parallelization enabled with 16 cores using the aforementioned processors. We used our own implementation of Lanczos in both cases, so our results are implementation-independent.

The main conclusion of this section is that our newly introduced Chebyshev approach is not only remarkably flexible, in the sense that it allows the study of generic (not necessarily autocorrelation) spectral functions, but also faster overall than the traditional Lanczos approach.

# 5 Concluding remarks

We studied the Kitaev-Heisenberg (K-H) and Kitaev-Ising (K-I) models on the honeycomb lattice for a 24-spin hexagonal cluster with periodic boundary conditions using three distinct approaches: Lanczos, TPQ and Chebyshev. This work is mainly divided in three parts. In the first part, we started by reproducing the results of Refs. [3–5] for the K-H model using Lanczos ED. Then, we recovered those results using microcanonical variants of the Lanczos (MCLM) and thermal pure quantum state (TPQ) methods and the Chebyshev polynomial Green's function (CPGF) method, independently of each other. For these three methods, we carefully examined the spectral and statistical convergence properties. While Lanczos is found to be ideal to approximate the ground state, we find that CPGF is the most efficient method capable of probing an arbitrary target energy with well controlled accuracy, proving to be faster than both MTPQ and MCLM on average throughout the phase diagram of the K-H model.

In the second part, we computed the temperature dependence of the N-N spin correlation, specific heat and entropy density. The aim of this study was to compare three methods based on Lanczos, TPQ and Chebyshev ideas, respectively: the finite temperature Lanczos method (FTLM), the canonical variant of TPQ (CTPQ) and the finite temperature Chebyshev Polyno-

mial method (FTCP), introduced in this paper. The MTPQ method is also considered because it is capable of estimating the temperature corresponding to each energy density remarkably accurately. Our implementations are bench-marked against the exponential tensor renormalization group results of Ref. [10]. We find our newly introduced FTCP method to be the most efficient and versatile of the three, namely showing a two-fold increase in speed compared with TPQ, while also avoiding the large low-temperature statistical fluctuations of FTLM.

The third and last part of this work started with the reproduction of Lanczos ED and TPQ results for the K-I model [61]. This bench-mark allowed us to validate our implementation and carry out a novel dynamical study for the K-I model, where we used a hybrid Lanczos-Chebyshev method. The latter was also shown to be more flexible and about 33% faster than Lanczos on average. Our detailed calculation of the dynamical spin susceptibility identifies signatures of the quantum phase transitions in the K-I model. This third part of our work also provides further evidence for the efficiency of FTCP, confirming the two-fold speed-up with respect to TPQ in finite temperature calculations for the K-I model.

In what follows, we summarize the key aspects that support the conclusions above for each part of our work.

All microcanonical methods show low statistical fluctuations and considering even just a single realization of the initial random state seems to suffice to achieve negligible deviations from the exact diagonalization results throughout the phase diagram of the K-H model. Unlike the number of realizations, the resolution plays a central role when comparing the performance of the three microcanonical approaches. In CPGF, finer resolutions always require more polynomials to achieve convergence. There is no one-to-one correspondence between the CPGF resolution and the effective resolutions of MTPQ and MCLM (which vary as more iterations are completed). Nonetheless, we managed to compare the three methods. TPQ showed an erratic convergence speed, with particularly significant slow-down for $\varphi = 0.506\pi$ in the K-H model, at which point around $4 \times 10^4$ polynomials/iterations are needed to achieve convergence and thus match the Lanczos ED results satisfactorily. On the other hand, CPGF requires very fine resolution to recover the ED results near the quantum phase transitions, thus converging relatively slower (but still faster than MTPQ) at these points of the phase diagram. Yet, away from quantum critical points, comparatively coarse resolutions are enough to reproduce the ED results. For most points of the phase diagram, CPGF has fast convergence and a relatively coarse input resolution suffices to match the results of ED. Even though the convergence behavior of MCLM throughout the phase diagram is not as predictable as CPGF, the convergence speed is comparable to that of CPGF and both typically converge faster than MTPQ. Overall, we find that CPGF requires less computer time and has well controlled accuracy through the resolution and number of polynomials, thus having a slight edge over MCLM. In spite of not being ideal for probing target energies with well controlled accuracy, we still find that MTPQ is very useful because of its ability to estimate the temperature corresponding to each energy density over the course of the iteration. This means that MTPQ is a viable method to carry out studies that would otherwise only be possible using canonical methods.

Regarding finite-temperature studies, we find shortcomings in both MTPQ and its canonical counterpart, CTPQ. Both seem advantageous at first sight due to their lower memory cost. However, in the case of MTPQ, this implies a trade-off that we show to greatly increase the computer time. On the other hand, in the case of CTPQ, a numerical instability limits its ability to probe very low temperatures. We show that the main competitor of TPQ, the Lanczos-based FTLM, has comparatively larger statistical fluctuations when studying the low-temperature behavior of the NN spin correlations. These fluctuations, along with a high number of matrix-vector multiplications per iteration in FTLM, limit the method's efficiency. Finally, we show that the newly introduced FTCP method circumvents the shortcomings of the other methods. Its statistical fluctuations are smaller than FTLM and comparable to the TPQ methods. Whilst the

FTCP memory cost is the same as FTLM (but slightly higher than TPQ), the trade-off is that the method is more efficient, i.e. twice as fast as MTPQ. This can be a crucial advantage in practical applications. Moreover, when using FTCP, accuracy can be controlled at each iteration, unlike in MTPQ, where the only way to guarantee sufficient accuracy is to increase the upper bound on the maximum eigenvalue of the spectrum by trial-and-error and carry out more costly, longer simulations. This is a demanding process, where one considers that accuracy is sufficient when no changes are detected in the relevant quantities for the desired temperature range as the upper bound is increased and the simulations are repeated. FTCP provides us with the the option of choosing the number of polynomials for convergence in each Chebyshev expansion throughout the iterative process, thereby ensuring that accuracy is maintained in the whole temperature range, without dramatically increasing the computational cost.

Our results show clear trade offs that must be taken into account when choosing which method to use. For example, MTPQ is designed to achieve maximum accuracy for the ground state. However, it cannot isolate excited states nor can it ensure uniform accuracy throughout large temperature windows. Concomitantly, the additional control afforded by the CPGF approach — which can access excited states directly — could be useful for studying non equilibrium systems, such as those studied in Ref. [9]. Another example is that of canonical methods. While we find Lanczos to be efficient for the computation of observables closely related to the Hamiltonian, it has large statistical fluctuations for more generic observables that might be of interest, such as the NN spin correlation in the K-H model.

A particularly powerful competing method is DMRG, originally devised to investigate 1D interacting systems [27–29]. DMRG aims to systematically truncate the exponentially large Hilbert space basis. The basis is rotated in the process in order to improve the accuracy of the truncation. This rotation is achieved via a series of global rotations generated by sweeping the lattice and thus focusing on a few sites at a time. The wide applicability of the DMRG procedure means that it can be used as a general purpose, sign problem free method. Moreover, it yields results that are competitive with QMC. However, the application of DMRG to 2D systems remains challenging [30]. The Chebyshev-based methods used throughout this paper are a potential alternative to DMRG because, unlike the latter, they pose no restrictions on boundary conditions and their accuracy can be precisely controlled by ensuring statistical convergence and, in the case of CPGF, by adjusting the spectral resolution. As far as the system size is concerned, Ref. [85] details the use of conservation laws to improve the efficiency of ED methods, particularly from the computer memory point of view. For example, translational symmetry implies that some configurations of the spins are equivalent up to a phase factor. This is a consequence of the block structure of the Hamiltonian, which gives forth to a reduced basis approach, enabling the study of larger system sizes. The only caveat is that systems with open boundary conditions and/or random couplings cannot be tackled with this approach.

To sum up, our results show that Chebyshev methods are more versatile and efficient than their Lanczos and TPQ counterparts, unless one is interested in properties that are well described using solely the ground state, or a small set of low-lying excitations. While in that case, Lanczos is still the method of choice, Chebyshev methods have significant advantages in various other scenarios, namely for the study of: properties that depend on an arbitrary target energy; finite temperature behavior of observables of interest that cannot be expressed in terms of the Hamiltonian and low-order polynomials of the latter; and dynamical quantities, such as spectral functions.

# Acknowledgments

This project was undertaken on the Viking Cluster, which is a high performance computing facility provided by the University of York. We are grateful for computational support from the University of York High Performance Computing service (Viking) and the Research Computing team.

**Funding information**    F. B. is supported by a DTP studentship funded by the Engineering and Physical Sciences Research Council. A.F. acknowledges the support from the Royal Society through a Royal Society University Research Fellowship.

# A  Stochastic trace evaluation

Statistical convergence is obtained when the error bars become acceptably small. This information is encoded in the scaling of the standard deviation with the number of used initial random states. In Fig. 13, we confirm that we obtain the expected scaling ($\sigma \propto N_{\text{rd.vec.}}^{-1/2}$) [49], that is our error bars can made as small as required by simply averaging over more realizations of the initial random state. This calculation was carried out with CPGF for a point in the phase diagram of the Kitaev-Heisenberg model with the parametrization of Ref. [3].

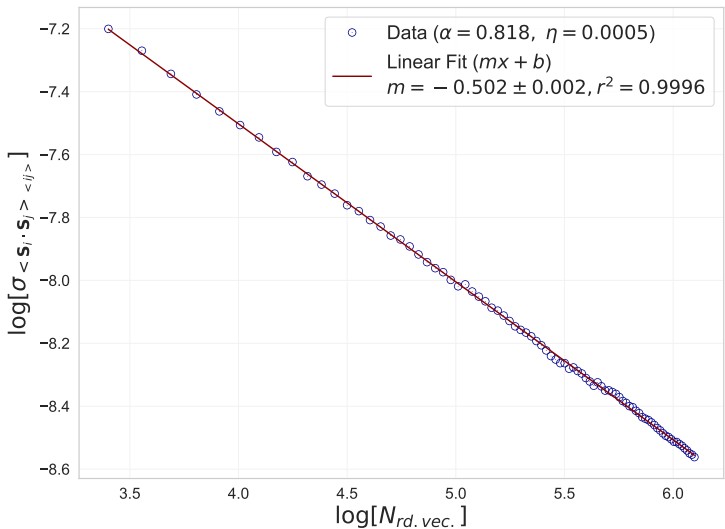

Figure 13: Behavior of the standard deviation of the NN spin correlation with the number of realizations of the initial random state for the CPGF method. Here, we consider a specific point of the phase diagram ($\alpha = 0.818$ in the parametrization of Ref. [3]). We obtain the expected scaling: $\sigma \propto N_{\text{rd.vec.}}^{-1/2}$.

# B  Spectral convergence and computational effort of CPGF

As mentioned in Sec. 4, the spectral convergence of CPGF follows a predictable pattern: the optimal number of polynomials needed for convergence, $N_{\text{poly}}^*$, is inversely proportional to the

resolution, $\eta$. Our results shown in the top panel of Fig. 14 confirm this behavior. In the CPGF, we used a stricter definition of convergence than before: A variation of less than $10^{-9}$ in the energy density between *three* consecutive iterations. This is necessary because convergence occurs in a "damped oscillatory" manner in CPGF (as we have shown in Fig. 7).

When targeting the ground state, the convergence of the CPGF method slows down close to critical points, despite showing comparably faster convergence in other parts of the phase diagram (center panel of Fig. 14). Surprisingly, MCLM behaves in a complementary fashion: convergence tends to become faster near a phase transition. Furthermore, each iteration is faster overall with CPGF, so even when in cases where both require comparable numbers of iterations for convergence, CPGF tends to be faster. Thus, the CPGF is faster than MCLM at reproducing the complete phase diagram.

The cost of targeting the excited states with CPGF is comparable to targeting the ground state. In contrast, MCLM requires significantly more iterations for convergence, resulting in a

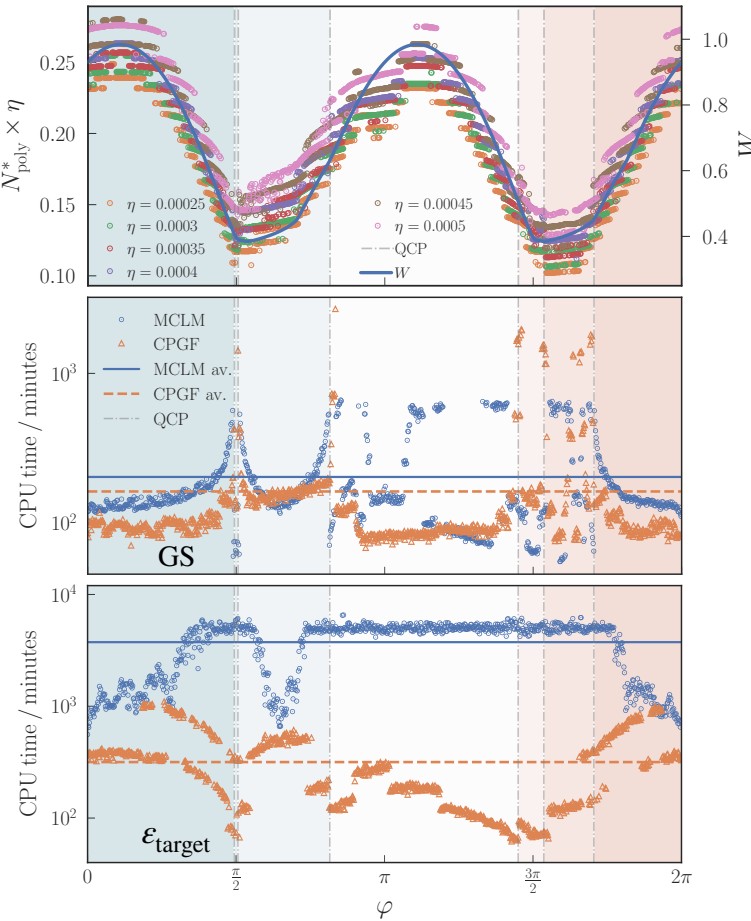

Figure 14: Top panel: Number of iterations required for convergence with CPGF times the required resolution (left vertical axis) and spectrum width (right vertical axis). $N^*_{\text{poly}}$ is approximately proportional to the spectrum width and inversely proportional to the required resolution. Thus, the number of iterations required for convergence can be estimated in advance, unlike in other approaches. Center panel: Computer time required for convergence using MCLM and CPGF to target the ground state. Bottom panel: Similar to the center panel, but now targeting excited states with energy $\varepsilon_{\text{target}}$. Here, we allowed a maximum of 10000 iterations with MCLM. In comparison, CPGF never required more than 3195 polynomials for convergence.

total CPU time about an order of magnitude larger than CPGF. The complementary behavior of the convergence properties of the two methods persists, i.e. the convergence speed of MCLM increases for the parts of the phase diagram where the convergence speed of CPGF decreases. Still, CPGF remains faster for the whole of the phase diagram when targeting the excited states. In conclusion, CPGF is the method of choice for targeting excited states.

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
