# Peer review of "Real-space spectral simulation of quantum spin models: Application to generalized Kitaev models"

_SciPost Physics Core, doi:SciPost Phys. Core 7, 006 (2024)_

## Round 1 · Referee Report · Anonymous (Referee 1) · 2021-10-29

Strengths

1) rigorous consistency checks of their method 2) well-written and clear structure of the manuscript

Weaknesses

1) Unclear, why this method is needed to compute ground state energies or ground state spin correlations 2) Only one application, the Kitaev-Heisenberg model 3) Too verbose in some parts 4) The method is already known 5) Comparison only to TPQ, no obvious difference

Report

The authors apply a Chebychev Polynomial method to study spectral properties of the Kitaev-Heisenberg model. The manuscript is well written, and the need for proper numerical methods is motivated well. The method is based on previous works, which have applied it to study spectral functions. Here, it appears as if the authors are using the method to study static quantities in the microcanonical ensemble, which is a valid application. The method is benchmarked against the TPQ method, in its microcanonical form, and Exact Diagonalization. The Kitaev-Heisenberg model is taken as a benchmark problem for the method. No new insight into its physics is added. There are several issues, that need to be addressed before the manscript can be considered for publication.

1) The method is only applied to show consistency with ground state Exact Diagonalization. While this is a good consistency check, this can clearly not be the purpose of this method. We see how ground state energy and ground state spin-spin correlations agree in Figs. 3 and 4. However, computing ground state expectation values are clearly better done using ED alone. When the authors go to N_{poly} = 40000 to achieve convergence of the spectral resolution, this is around 200 times more expensive than a simple ground state computation. The method would need to be compared at finite energies, where it is a valid application. The paper cannot be published if it only shows consistency checks at T=0.

2) The use of the microcanonical ensemble is poorly motivated. It would be good to explain, where the advantages lie in this approach and compare this to the canonical TPQ (Sugiura, Shimizu, Phys. Rev. Lett. 111, 010401 (2013)) or the finite-temperature Lanczos method (see e.g. Jaklic, Prelovsek, Phys. Rev. B 49, 5065(R) (1994))

3) A well-known method to study spectral properties in the microcanonical ensemble is the microcanonical Lanczos algorithm, see e.g. Long, Prelovšek, Shawish, Karadamoglou, Zotos, Phys. Rev. B 68, 235106 (2003) or https://www.cond-mat.de/events/correl17/manuscripts/prelovsek.pdf. A comparison to this method would greatly improve the manuscript.

4) Static observables in the microcanonical ensemble are somewhat interesting, but since previous studies have already applied this method to spectral functions, it would be good, if the authors could also attempt this. This has already been done in the paper that is cited by the authors: Braun, Schmitteckert, Phys. Rev. B 90, 165112 (2014), but applying it to the Kitaev Heisenberg model would be a novel step.

5) It is not clear from their manuscript whether the Chebychev approach is advantageous to TPQ. The authors need to compare results at finite energy, where Exact Diagonalization with e.g. a simple Lanczos algorithm is not feasible anymore.

6) Since the Heisenberg-Kitaev model is only used as a benchmark, the description of its physics can be shortened. After all, there are no new insights added to the understanding of this model.

7) Since this paper is benchmarking the Chebychev method, it would be advisable to not just have one model it is applied to, but rather two or three.

While both the method and the application are interesting, the above-stated problems hinder me from recommending this manuscript for publication in its present form. However, upon addressing these major points, the manuscript might be suitable for publication in SciPost.

---

## Round 1 · Referee Report · Anonymous (Referee 2) · 2021-11-9

Strengths

1-The authors provide a combined benchmark of the TPQ and CPGF approach for a specific frustrated quantum spin model.
2-The methods are applied here to a relevant and timely quantum spin model: the Kitaev-Heisenberg model.

Weaknesses

1-No new physics results are reported.
2-The considered methods have been presented in detail in earlier publications.
3-No comparison is performed for models for which, e.g., QMC or iPEPS results are available, which would allow the authors to test the proposed method for larger systems sizes than those available to ED.
4-There are several minor deficits in the presentation, see list below.

Report

The authors perform a systematic benchmark and comparison of the TPQ and CPGF method as applied to a specific frustrated quantum spin model, the Kitaev-Heisenberg model. This type of analysis is certainly interesting and timely, and the authors also address a timely and relevant frustrated quantum spin model.

I request several changes, as specified below, to be performed on the manuscript, before approving that the general acceptance criteria for SciPost Physics are met by this manuscript. Given that no new physics results are reported here and the considered methods have in fact been put forward (applied to different models) already in earlier publications, I don't find one of the expectations for a publication in SciPost Physics to be particularly met by this submission.

Requested changes

1-The sentence in the introduction, "The severity of the problem depends on the nature of the correlations in the model." is not correct: it is a well known fact and by now documented by several example cases, that the severeness of the sign-problem depends on the computational basis used in a QMC approach. The quoted statement thus needs to be corrected and augmented by appropriate references.
2-The abbreviation "STE" at the bottom of page 4 needs to be introduced. Also not defined at the point of first usage explicitly are $N_{rd.vec.}$, $D$.
3-It is not clear what is meant by the $xy-plane$ on page 5.
4-The meaning of the coefficients ${c_i}$ on page 6 is not clear.
5-Please provide the range $n=0,1,2,...$ when the Chebyshev polynomials are first defined (after eq. (5)).
6-The bond lines in Fig. 2 are hardly seen, the arrows are too long. In later figures, the boundary boxes are too faint and the font sizes too small.
7-At the end of the introduction to Sec. 4 please motivate, why the n.n. correlations will be considered only. What about longer ranged correlations?
8-Please reverse the order in the legend in Fig. 6, for easier readability.
9-While DMRG is discussed as an competing method, the authors should also mention other tensor-network-based approaches, such as iPEPS, where remarkable advances have been demonstrated recently in exploring the thermodynamics of frustrated 2D quantum spin systems.
10-Extend the current analysis to additional models, in particular such models, for which larger-scale benchmark data is available, i.e., for system sizes that extend beyond those accessible to ED.

---

## Round 1 · Referee Report · Anonymous (Referee 3) · 2021-11-11

Strengths

  1. This paper tries to show the numerical reliability of the Chebyshev Polynomial Green Function (CPGF) to investigate the ground-state properties of the Kitaev-Heisenberg model.

  2. This paper shows that the CPGF method can detect the signature of the ground-state phase transition in the Kitaev-Heisenberg model with comparable accuracy as the results obtained by the conventional exact diagonalization method.

  3. The paper also points out that the CPGF method can be used to access even the excited energy states, which are useful for studying the non-equilibrium systems.

Weaknesses

  1. Clear advantage is not clarified in using the CPGF method instead of the conventional Lanczos method to investigate the ground-state properties of the quantum frustrated magnets although the paper investigates the numerical reliability only in the ground-state of the Kitaev-Heisenberg model.

  2. The paper recommends using the TPQ method to get the ground-state energy instead of the ED method. However, the clear advantage is not clarified to use the TPQ method in terms of the required memory cost, the needed number of (TPQ/Lanczos) iteration, and the number of initial random vectors.

  3. The numerical results in Fig. 6, the left and right panel data should match in the end, but not so. This inconsistency easily violates the reliability of the CPGF method, but the paper does not explain the reason.

Report

The current paper investigates mainly the reliability and usefulness of the Chebyshev Polynomial Green Function (CPGF) to study the ground-state properties of the Kitaev-Heisenberg model as an alternative numerical method to the quantum Monte Carlo (QMC) method and the exact diagonalization (ED) method. By careful comparison to numerical data by the conventional ED Lanczos method and the thermal pure quantum state (TPQ) method, some numerical reliabilities are clarified, and the computational costs in the CPGF method are also partly reported. However, the current manuscript does not seem to meet the acceptance criteria of the SciPost Physics because of the lack of the clear advantage in using the CPGF method instead of the ED method.

To improve the paper further, I would like to authors to comment on the following points.

  1. Please clarify the computational memory costs in the CPGF, TPQ, and Lanczos methods to compute the energy and the correlation functions in the ground state. I think these methods require the almost same memory cost ($\sim$ 2 or 3 $D$). But the TPQ and the CPGF methods require few initial random vectors to get the same level of accuracy as the results obtained by the ED Lanczos method.

The authors may answer that only a few or 1 realization is enough to use the TPQ method, but that is the case in larger enough systems because the error scales as $\sim 1/\sqrt{D}$. (Can we believe this scale also in the CPGF method?)

The Lanczos method does not need to take the average over the initial random vectors and does not require $\sim$ $10^4$ iteration (typically, $10^2$~$10^3$ in the Lanczos method).

From the viewpoint of the memory costs, the number of initial realizations, and the number of iteration, for me, the conventional Lanczos method is the best way among these three methods to investigate the ground state physics.

In the conclusion of this manuscript, the authors suggest a hybrid TPQ-CPGF approach. However, based on the above reasons, the only ED Lanczos approach seems to be better for the computational costs to study the ground-state physics.

And it is not also shown that the TPQ-CPGF approach is better than the ED-CPGF approach to study the microcanonical excited state physics.

  1. To use the TPQ and CPGF methods, not only the lowest energy $E_{\rm min}$ but also the maximum energy $E_{\rm max}$ are needed in advance. The authors suggest using the ED or the TPQ method in getting the lowest energy state, but the general way to get the $E_{\rm max}$ is not written. Of course, in some simple cases, we can know the value easily. For example, in the simple antiferromagnet, we can use the energy of the fully polarized state as the $E_{\rm max}$. But in general, could we avoid using the Lanczos-like method to know the $E_{\rm max}$?

  2. Please explain explicitly the inconsistency between the left and right panels in Fig. 6. These calculation results should match in the same parameter $\alpha$ in the large enough iteration. And please use the same scale in the vertical axis to make the comparison easier for readers.

  3. The authors say "CPGF is a potential alternative to DMRG because it poses no restrictions on dimensionality and its accuracy can be precisely controlled by ensuring statistical convergence and by adjusting the spectral resolution." on page 13. However, I think that the current CPGF method is also suffered from the huge memory cost which increases exponentially as in the Lanczos and the TPQ methods. I don't think that the current CPGF method can be used in three-dimensional systems from this point.

  4. When we have the degeneracy in the ground states or the excited states, can we use the CPGF method as a reliable method? There are often degenerated ground states and excited states in the quantum frustrated systems.

---

## Round 2 · Referee Report · Anonymous · 2023-12-11

Report

The authors provide a substantially revised, extended and improved version of their manuscript. In particular, they now introduce two new methods, the finite temperature Chebyshev polynomial (FTCP) method for computations of canonical expectation values and a hybrid Lanczos-Chebyshev (HLC) method to compute spectral functions. The authors provide a detailed analysis of the performance of the various methods and highlight subtle advantages. I find some of their discussion, such as the energy-cutoff-dependence of the accuracy of the MTPQ particularly interesting. The authors also extended substantially on the physics side, and in particular now provide some dynamical data for the K-I model. Unfortunately, they do not go beyond the system with 24 sites. Nevertheless, I think that the paper may now be considered appropriate for SciPost Physics, but I would still request a few minor changes, as given below.

Requested changes

-1- Address the limitations on system size more explicitly: which new steps could be taken to go well beyond those feasible thus far with your methods?
-2- Explain the meaning of the white space in Fig. 12 or adapt the the color bar, in case that white means 0.
-3- I do not find evidence in the paper regarding the second part of the statement "While Lanczos I found...", the last sentence of the first paragraph in Sec. 5. If you want to keep this statement, please provide explicit data for excited state targeting.
-4- The list of references regarding the basis-dependence of the sign in QMC is incomplete. Cite also: T. Nakamura, Phys. Rev. B 57, R3197 (1998); A. Honecker, S. Wessel, R. Kerkdyk, T. Pruschke, F. Mila, and B. Normand, Phys. Rev. B 93, 054408 (2016); F. Alet, K. Damle, and S. Pujari, Phys. Rev. Lett. 117, 197203 (2016); L. Weber, A. Honecker, B. Normand, P. Corboz, F. Mila, and S. Wessel, SciPost Phys. 12, 54 (2022).

---

## Round 2 · Referee Report · Anonymous · 2023-12-15

Strengths

1. This paper introduced several well-known non-biased numerical (ED-like) methods in pedagogical manner that have the potential to surpass the Lanczos method to investigate excited states and finite-temperature physics.

2. This paper developed three methods (CPGF/FTCP/HLC methods) based on Chebyshev technique which can be superior to the previous methods in some aspects (such as in computational time and in accuracy).

3. Using HLC method, the paper found the signatures of quantum phase transition in a Kitaev-Ising model via spin dynamics.

Weaknesses

1. This paper did not check the performance of the CPGF method for excited states even though this method was developed for excited state as an alternative method to the traditional Lanczos method.

2. There may be inconsistency between the computational results of specific heat and entropy obtained by CTPQ method.

3. No explicit reason was explained why the FTCP method can avoid convergence problem at lower temperature regime.

Report

First of all, I would like to express a respect for the author's two years of effort. Compared to their 1st paper from two years ago, this paper begins with an pedagogical introduction to commonly used non-biased methods in this field. This paper also then provides a wealth of numerical data demonstrating the merits of improvements based on the Chebyshev method. Additionally, the paper successfully calculates spin dynamics using the HLC method and captures the signatures of quantum phase transitions in a Kitaev model. While there may not be groundbreaking discoveries in physics, I believe that the series of methods presented in this paper hold value for publication as they have potential applications to other models, making it a worthwhile methodological contribution.

Requested changes

1. It would be better to write the reason why the FTCP method can avoid the problem to evaluate the specific heat (in figure 9) in contrast to the CTPQ method. I guess, the reason is because you use \tilde{h} in stead of h.

2. It would be better to cite some references for your FTCP method. In your FTCP method, you use Chebyshev polynomial expansion instead of Taylor expansion to express \exp{-\beta/2 h} in the CTPQ method. There are several references which already used Chebyshev polynomial expansion in the CTPQ method such as, https://arxiv.org/abs/1503.06111v1, https://journals.aps.org/prl/abstract/10.1103/PhysRevLett.121.220601. Of course, these papers didn't show the method details and benchmark as you showed. But it would be nice to cite their papers, and if there were difference in their methods and your method, you can also write the difference in your paper.

3. Could you explain why you chose the ground state to investigate the benchmark of the CPGF method (in Fig.6)?
I think for the ground state, the traditional Lanczos method is the best. I also think that some readers are interested in if the FCPT method is better than the MCLM method also to see the excited states.

---

## Round 2 · Author Response

Expert Referee Report 1

We kindly thank the referee for carefully reading our manuscript, pointing out its strengths and shortcomings, and providing valuable suggestions for improvement. Please note that in the original submission, the convergence speed and performance of CPGF were severely underestimated due to a mistake in the input of the spectrum bounds. This has now been corrected, leading to a major boost in computational performance. We apologise for this mistake. We also now consider a broader parameterization of the Kitaev-Heisenberg model as employed by Chaloupka et al. Phys. Rev. Lett. 110(9), 097204 (2013). We are confident that we addressed the 5 mentioned weaknesses in this resubmission.

First, we reply to the general points made by the Referee (numbered and in quotes):

1) “Unclear, why this method is needed to compute ground state energies or ground state spin correlations”

We agree with the referee: CPGF is not needed to compute ground state energies or spin correlations. The advantage of the CPGF method is in probing arbitrary target energies that need not be close to the ground state (i.e. target energies beyond low lying excitations that are accessible using the standard Lanczos method). Notice that in this resubmission, we also compare CPGF with the microcanonical Lanczos method (MCLM), which is also capable of probing arbitrary target energies. We illustrate CPGF by considering the ground state energy as the target so that we can benchmark it against Lanczos exact diagonalization (ED) and the microcanonical variant of TPQ (which we refer to as MTPQ, as opposed to the canonical variant, referred to as CTPQ). Please note that the MTPQ recursion achieves its maximum accuracy as the ground state is approached, so the latter is a natural target energy to compare all methods. As noted by the referee, another reason to consider the ground state is implementation validation. Since the ground state Lanczos results are well established, matching between CPGF and ED validates our implementation, serving as a consistency check. Notwithstanding, our resubmission goes beyond a simple consistency check, by showing that CPGF compares favourably with both MCLM and MTPQ. Another important remark is that on this resubmission we also introduce two methods that were not included in the original submission: the finite temperature Chebyshev polynomial (FTCP) method for the evaluation of canonical expectations and a hybrid Lanczos-Chebyshev (HLC) method to compute spectral functions. We believe that the development of these methods is a significant contribution. In particular, while ED remains the go-to method for ground-state (T = 0) physics, we give compelling arguments in favour of FTCP for finite-temperature calculations.

2) “Only one application, the Kitaev-Heisenberg model”

We added two subsections, where we apply the newly developed FTCP and HLC approaches to the Kitaev-Ising (K-I) model. Specifically, we use FTCP to study the temperature dependence of relevant quantities, such as the specific heat and the entropy density. We use HLC to study the dynamical spin susceptibility of the K-I model. Lanczos ED results for this model are also reproduced in order to validate our implementation.

3) “Too verbose in some parts”

The revised manuscript uses a more concise and easier-to-understand writing style. Following the referee’s suggestion, section 4 (“Applications”) refrains from discussing the physics of the models and focuses solely on the methods and their advantages and challenges. The exception is Sec. 4.2.3 that introduces new results, for which we provide a physical interpretation.

4) “The method is already known” and 5) “Comparison only to TPQ, no obvious difference”

Even though CPGF was already known prior to this submission, it had never been applied to interacting quantum spin systems. We believe that this resubmission makes it clearer that it has interesting advantages with respect to state-of-the-art methods for strongly correlated systems, namely MCLM and TPQ. On this revised version of the manuscript, we also introduce two methods that we developed while working on the points previously made by the referees. Both of these methods show clear strengths. On the one hand, FTCP is found to be two times faster than microcanonical TPQ (MTPQ) and is immune to both the numerical instabilities of canonical TPQ (CTPQ) and the low-temperature fluctuations that plague FTLM. On the other hand, the newly developed HLC method for spectral functions is more flexible than the continued fraction Lanczos approach and is shown to be 33% faster on average in our extensive tests (about 4000 independent simulations). In this resubmission, we extend the comparison far beyond just the microcanonical variant of TPQ. We now compare our methods to the microcanonical and canonical variants of TPQ, MCLM, the Finite Temperature Lanczos method (FTLM) and the Lanczos method using the continued fraction approach to compute spectral functions.

Regarding the specific issues brought up by the referee, we believe that we have resolved every item:

Items 1) and 2)

We added Sections 4.1. 2 and 4.2.2., where we show that FTCP compares favourably with the other methods. Its memory cost is similar to FTLM, but it avoids low-temperature statistical fluctuations. While MTPQ also avoids these fluctuations while having 50% lower memory cost, FTCP uses half the computer time of MTPQ thanks to its better convergence properties and the accuracy control it offers. Unfortunately, CTPQ has numerical stability issues at the very low temperatures that are of particular interest in these systems. For these systems, the canonical ensemble is typically more useful. Following the referee’s suggestion, we now include comparisons with the canonical TPQ and finite temperature Lanczos methods. We find that the advantage of the microcanonical ensemble is mainly in MTPQ. Since this method is able to estimate the actual temperature corresponding to each iteration, one can actually compute the temperature dependence of relevant quantities using MTPQ. Unlike CTPQ, MTPQ does not suffer from numerical instabilities at low temperatures. However, this comes at a high computational cost. Moreover, the accuracy increases as more iterations are completed, which in practice means that MTPQ’s high-temperature results are unavoidably less reliable. Our newly introduced method (FTCP) solves both of these problems.

Item 3)

We compare CPGF with MCLM, and find that CPGF is more efficient on average. Moreover, the resolution is given as an input to CPGF. The scaling of the number of polynomials needed for convergence with resolution is known a priori in CPGF. We confirm this expected behaviour in Fig. 14. The advantage of the added control over resolution in CPGF is that one can predict how many iterations will be required to reach a certain accuracy. This is not known a priori in MCLM.

Item 4)

We agree with the referee on this point. Applying Chebyshev-based methods to spectral functions has proven fruitful in the past and they can certainly be used for this class of systems. We found that spectral functions had already been computed with Lanczos for the Kitaev-Heisenberg model in Phys. Rev. B 95, 024426 (2017). So, we applied a hybrid Lanczos-Chebyshev (HLC) method to recover results of this paper and discovered that our Chebyshev-based method has some advantages in terms of efficiency and flexibility. Additionally, we took another suggestion of the referee into account and applied this method to another model. In Sec. 4.2.2., we study the dynamical spin susceptibility of the Kitaev-Ising model for the first time to our knowledge and find dynamical signatures of the quantum phase transitions first described in Phys. Rev. Lett. 118, 137203 (2017).

Item 5)

We find that FTCP is advantageous with respect to MTPQ and CTPQ. Similarly to MTPQ, it is immune to the low-temperature numerical instability of CTPQ. Yet, it is 2 times faster than MTPQ and has better control over the accuracy of the results. This is because the accuracy of MTPQ improves for low temperatures, while in FTCP the accuracy depends only on the number of polynomials used on the Chebyshev expansion at each inverse temperature step (see Eq. (35)).

Item 6)

We agree that our work does not produce new insights into the physics of the Kitaev-Heisenberg model. This model is chosen to benchmark our Chebyshev-based methods. We do introduce some insight into the physics of the Kitaev-Ising model, in the revised version. These models are well described, respectively, in Phys. Rev. B 95, 024426 (2017) and Phys. Rev. Lett. 118, 137203 (2017). So, we keep our descriptions to a minimum and only discuss these models to explain the conventions we used to parametrize energy scales.

Item 7)

We now apply the Chebyshev methods to the Kitaev-Heisenberg (Sec. 4.1) and Kitaev-Ising models (Sec. 4.2).

Expert Referee Report 2

We kindly thank the referee for carefully reading our manuscript, pointing out its strengths and shortcomings, and providing valuable suggestions for improvement. Please note that in the original submission, the convergence speed and performance of CPGF were severely underestimated due to a mistake in the input of the spectrum bounds. This has now been corrected, leading to a major boost in computational performance. We apologise for this mistake. We also now consider a broader parameterization of the Kitaev-Heisenberg model as employed by Chaloupka et al. Phys. Rev. Lett. 110(9), 097204 (2013).

First, we reply to the general points made by the Referee (numbered and in quotes):

1) “No new physics results are reported.”

In this resubmission, we report new results, i.e. dynamical signatures of the quantum phase transitions in the Kitaev-Ising model (see Phys. Rev. Lett. 118, 137203 (2017)).

2) “The considered methods have been presented in detail in earlier publications.”

We introduce two novel methods that were not included in the original submission: the finite temperature Chebyshev polynomial (FTCP) method for computations of canonical expectations and a hybrid Lanczos-Chebyshev (HLC) method to compute spectral functions. We believe that the introduction of these novel methods is a significant contribution.

3) “No comparison is performed for models for which, e.g., QMC or iPEPS results are available, which would allow the authors to test the proposed method for larger systems sizes than those available to ED.”

In our current implementation, these Chebyshev methods cannot reach the system sizes of QMC or tensor networks. The limitation is the relatively high memory cost of these Chebyshev methods, which is approximately the same as Lanczos ED applied to the ground state. Both Chebyshev methods and “ground state” Lanczos ED allow larger systems than full exact diagonalization, i.e. the brute force approach, where all eigenvalues and eigenvectors are obtained. This is because their memory requirements are much smaller than full ED. Lanczos, TPQ and Chebyshev all share the following feature. When carrying out matrix-vector multiplications involving the Hamiltonian “on-the-fly”, which is the most memory-efficient technique that is currently available, one still has to store a few vectors of the size of the Hilbert space. In this regard, QMC has a clear advantage because it relies on the concept of importance sampling to probe only the relevant part of the Hilbert space, and thus bypasses the need to store such large vectors in memory. The advantage of the Chebyshev methods with respect to Lanczos or TPQ-based methods is better control of accuracy, more flexibility and higher efficiency. Chebyshev methods are also generally applicable, sign-problem-free and work equally well, irrespective of the number of spatial dimensions, model complexity and type of boundary conditions.

4) “There are several minor deficits in the presentation, see list below.”

We believe that we have addressed all the minor deficits in our presentation, as detailed below.

We went through the list of requested changes and included them in our resubmission. Below, we point the referee to the location of these changes in the manuscript:

1) We correct the statement about QMC in page 3: “ The severity of the problem depends on the computational basis used to tackle the specific model” and provided more references about this topic: Annual Review of Condensed Matter Physics 10 (1), 337 (2019), Phys. Rev. Lett. 126(21), 216401 (2021), Phys. Rev. B 105 (16), 165124 (2022), Phys. Rev. E 99 (3),033306 (2019) and Phys. Rev. B 105(19), 195130 (2022).

2) We introduce the abbreviation “STE” in page 5: “This technique, dubbed stochastic trace evaluation (STE), is ubiquitous in the study of condensed phases and is used in ED methods…”; we also define the other quantities, respectively in “The rationale in the STE is to approximate the trace of an operator by an average of expectation values using $N_\mathrm{rd.vec.}$ random vectors” and “The relative error scales favorably with the Hilbert space dimension, $D$”.

3) We now clarify the meaning of these variables on footnote 3.

4) The coefficients are now defined on footnote 2.

5) We now define the range of the Chebyshev polynomials in page 10, after the sentence “As customary, we work with Chebyshev polynomials of the first kind…”

6) We improved the readability of the figures throughout the manuscript.

7) We added a sentence, motivating the use of n.n. correlations in page 19: “The NN spin–spin correlation is used for our bench-mark for two reasons. Firstly, in Phys. Rev. B 95, 024426 (2017), the authors show that longer-range correlations vanish in the vicinity of the spin liquid phases. Given that we are particularly interested in this region of the phase diagram, it is reasonable to focus on NN correlations. Secondly, the step-like behavior of the NN spin--spin correlation coincides with quantum critical points (see Phys. Rev. B 95, 024426 (2017) ). Moreover, the behavior of the NN correlation with temperature is intimately connected to peaks in the specific heat (see Phys. Rev. Research 2 (4), 043015 (2020) ) — that we also compute using Lanczos, TPQ and Chebyshev --- and that are particularly relevant experimentally (see Phys. Rev. B 99(9),094415 (2019) ).”

8) The data shown in Fig. 6 of the original manuscript was re-obtained in the context of the broader parameterization of the Kitaev-Heisenberg model in Phys. Rev. Lett. 110(9), 097204 (2013). In the new revised manuscript the analogous comparisons are contained in figure 5 and 7. In these figures, we improved the readability by having the same scales on the left and right panels and by using the same colour scheme for the range of parameters of the model where the methods are being compared.

9) We mention Phys. Rev. Research 2 (4), 043015 (2020), where the authors use an exponential tensor network approach to study the Kitaev model, and show that our results match those of this approach in Sec. 4.1.2. iPEPS is also mentioned in page 3, with some extra references added.

10) As mentioned above, unfortunately the current (also the first) implementation of our methods to quantum spin models do not go beyond the system size limitations of ED because of their similar memory cost. We compared our results for the Kitaev model with Phys. Rev. Research 2 (4), 043015 (2020) and confirmed that they agree with their larger-scale bench-mark data. We also applied our newly introduced HLC to compute the dynamical spin susceptibility of the Kitaev-Ising model and put forward new results concerning these dynamical signatures of the quantum phase transitions in this model.

Expert Referee Report 3

We kindly thank the referee for carefully reading our manuscript, pointing out its strengths and shortcomings and requesting the improvement of important points that improved our manuscript considerably. Please note that in the original submission, the convergence speed and performance of CPGF were severely underestimated due to a mistake in the input of the spectrum bounds. This has now been corrected, leading to a major boost in computational performance. We apologise for this mistake. We also now consider a broader parameterization of the Kitaev-Heisenberg model as employed by Chaloupka et al. Phys. Rev. Lett. 110(9), 097204 (2013).

We are confident that we satisfactorily addressed all the 5 points mentioned by the referee in this resubmission:

1) In this resubmission, we introduce two methods that were not included in the original submission: the Finite Temperature Chebyshev Polynomial (FTCP) method and the hybrid Lanczos-Chebyshev (HLC) method to compute spectral functions. We believe that the introduction of these novel methods is a significant contribution. In particular, while Lanczos remains the go-to method for the ground state, we give compelling arguments as to why it might be preferable to use FTCP when one desires to go beyond T=0.

The computational costs are indeed those mentioned by the referee. Specifically, memory-wise, the dominant contributions are O(2D) for the TPQ methods and O(3D) for the Lanczos and Chebyshev-based methods. So, at the first sight, all other factors being the same, TPQ may seem preferable to go beyond T=0. However, in this updated version of our paper, we show that this is not actually the case. While the memory cost of the FTCP is about 50% higher than TPQ (that is, O(3D) instead of O(2D)), FTCP is 2 times faster than TPQ in terms of computer time. We attribute this to the better convergence properties of the Chebyshev expansion and the modest number of operations per iteration that is required in Chebyshev. In conclusion, while we agree with the referee in saying that Lanczos is superior for T=0, in this updated version of the manuscript, we show that Chebyshev-based methods can be advantageous for studies of: target energies away from the ground state and low-lying excitations, temperature dependence of experimentally relevant quantities such as spin correlations, specific heat and entropy, and dynamics, for example the dynamical spin susceptibility, studied in Sec. 4.2.3. In this last section, we present novel results, showing dynamical signatures of the quantum phase transitions in the Kitaev-Ising model (see Phys. Rev. Lett. 118, 137203 (2017)).

2) In general, the most reliable method to obtain the energy bounds is Lanczos. For the specific case of the Kitaev-Heisenberg model, we noticed that a useful symmetry allows one to avoid the computation of the maximum energy (see Sec. 4.1.1.). This is because there is a mapping between minimum and maximum energies for different values of the parameter of the Kitaev-Heisenberg model.

3) We improved the readability of the figures overall. Namely, we used the same scale in all of these comparisons (see figures 5 and 7). The inconsistency that was pointed out by the referee was due to the fact that the original CPGF simulations were not fully converged, i.e. a yet finer resolution was needed. In the revised version, we show that the MTPQ and CPGF results do match (see right panels of figures 5 and 7). In order to ensure that CPGF was fully converged, we systematically considered finer resolutions until the results were resolution-independent, whilst increasing the number of iterations accordingly. Then, we compared the coarsest resolution CPGF result that had already converged with TPQ.

4) We realised that this sentence was misleading. Indeed, like the referee pointed out, CPGF has similar memory cost to Lanczos and TPQ. What we meant was that CPGF and the other Chebyshev methods do not require any specific boundary conditions as the dimensionality is varied. For example, in the case considered in our work, we considered a 2D hexagonal cluster with periodic boundary conditions. DMRG typically considers open boundary conditions along at least one of the spatial dimensions. We changed the sentence in question to: “The Chebyshev-based methods used throughout this paper are a potential alternative to DMRG because, unlike the latter, they pose no restrictions on boundary conditions and their accuracy can be precisely controlled by ensuring statistical convergence and, in the case of CPGF, by adjusting the spectral resolution.” (page 33)

5) CPGF considers target energies contained within a (uniform, well-defined) resolution. Thus, the state that results from the iterative process will always be a linear combination of the degenerate states in question. This applies to Lanczos and TPQ as well. Unfortunately, Chebyshev-based methods do not have any obvious advantages over Lanczos and TPQ as far as degeneracies are concerned.

In conclusion, we believe that in the revised manuscript, we have shown that while Lanczos ED remains the preferable method for ground state studies, Chebyshev methods can be advantageous for other applications, namely: a) studying target energies, in which case we found CPGF to be more efficient than the micro canonical Lanczos method (MCLM) and TPQ; b) finite temperature studies, where the finite temperature Lanczos method (FTLM) and TPQ have shortcomings, respectively, large low-temperature fluctuations and two-fold increased computer time compared with FTCP; c) computing spectral functions, in which case we considered a particular range of parameters in the Kitaev-Heisenberg model, performed 4000 independent simulations and found that on average, our newly introduced hybrid Lanczos-Chebyshev method was 33% faster than the continued fraction Lanczos approach. Lastly, we carried out a novel calculation of the dynamical spin susceptibility for the Kitaev-Ising model and found dynamical signatures of the quantum phase transitions in this model. We provided a brief physical interpretation for these results. All summed up, we believe that Chebyshev-based methods can be advantageous with respect to both Lanczos and TPQ-based methods, except to obtain the ground state, in which case Lanczos is the best method. In particular, FTCP has a clear advantage over the state of the art method for finite temperature studies (TPQ), achieving the same results 2 times faster, as shown in Sections 4. 1. 2. and 4. 2. 2., with only a minor increase in memory cost mentioned on the first point of this reply.

---

## Round 2 · List of Changes

- The whole paper has been re-written, including the abstract. The title also had a minor change to reflect the inclusion of another model in our study. These changes were made in order to accommodate all the suggestions made by the referees. The underlying structure has been kept, but there are more subsections in Sec. 3, where the additional considered methods are discussed. Crucially, Sec. 4 suffered a major overhaul, with Sec. 4. 2. added. Also, Sec. 4.1.2 is completely new. Changes to Secs. 1 and 2 are relatively minor.
- In the revised manuscript, all figures are completely new, except for Figure 1 and Figure 13. This is because we used a broader parametrization of the Kitaev-Heisenberg model in the revised version. Figures 3-7 of the new manuscript contain results that are analogous to those of figures 4-6 of the old manuscript, with some additional material.
- Figures 8 and 9 show the results of an extensive comparison of finite temperature methods for the Kitaev-Heisenberg model.
- Figure 10 is a consistency check for our implementation of the methods for the Kitaev-Ising model. Figure 11 presents a comparison between two particularly relevant finite temperature methods in the context of the Kitaev-Ising model. Figure 12 displays novel dynamical signatures of the quantum phase transitions in the Kitaev-Ising model.
- Sec. 5 has been extensively updated to include the conclusions of our additional studies of finite temperature and dynamics.
- Appendix B has been added. This serves the purpose of illustrating important advantages of the CPGF method that were not apparent in the original manuscript.

---

## Round 3 · Referee Report · Anonymous (Referee 3) · 2023-12-28

Report

This revised manuscript gives a lot of useful contents for ED-like methods and meets acceptance criteria.

---

## Round 3 · Referee Report · Anonymous (Referee 2) · 2024-1-8

Report

In the second revised manuscript the authors responded appropriately to my further remarks.

---

## Round 3 · Author Response

Reply to Referee Report 1

We are pleased that the referee recognizes the improvements in the extended revision of the original manuscript and, in particular, appreciated our detailed analysis of the performance of different methods, the introduction of two new methods (FTCP and HLC), and the expansion on the physics side. We are also glad that the referee showed a particular interest in the energy cutoff dependence of TPQ. We believe that we have implemented the minor changes requested by the referee in this second re-submission:

  1. The limitation on system size stems from the computer memory requirements for the storage of D-vectors, i.e. vectors with the dimension of the Hilbert space; this limitation applies to all the methods discussed in our work (Lanczos, TPQ and Chebyshev-based). Yet, in AIP Conf.Proc.1297:135,2010, the author explains how conservation laws can be used to improve the efficiency of Lanczos exact diagonalization algorithms, in particular from a computer memory standpoint. The arguments presented in the aforementioned reference apply to the Chebyshev approach as well (and to TPQ, for that matter). For example, by using translational symmetry, some configurations of the spins are seen to be equivalent to one another, apart from a phase factor. Implementing this symmetry explicitly in our code (via a reduced basis that reflects the block structure of the Hamiltonian) would enable the study of larger systems. Here, there is an important caveat: this technique does not apply to more general systems, for example with open boundary conditions and/or random couplings. A sentence was added to the paper explaining this technique in the second to last paragraph of Sec. 5: “As far as the system size is concerned…”

  2. We improved the color bar in Fig. 12, so as to clarify the transition to white. We have also added a sentence in the caption explaining that white space indeed corresponds to a vanishing spin susceptibility, as correctly inferred by the referee.

  3. We extended our discussion beyond the ground state, presenting data to back our claim in the last sentence of the first paragraph in Sec. 5. We target excited states (now defined in Fig. 3 in the revised version of the manuscript) and compute the nearest neighbor spin-spin correlation for these states as well (see bottom panel of Fig. 4 in the revised version). Although there is already a difference in performance for the ground state, we find it to be more pronounced for excited states, i.e. CPGF is about an order of magnitude faster than MCLM when the excited states are targeted. We attribute this to the uniform convergence of the CPGF, which guarantees that the required Chebyshev iterations remain broadly the same, regardless of the target energy. Differences in convergence speed throughout the phase diagram are explained by the fact that the resolution must be adjusted in order to probe each excited state with sufficient resolution to compute the spin correlator accurately. In turn, finer resolutions require more polynomials.

  4. We augmented our list of references with the suggestions of the referee (see page 3).

Reply to Referee Report 2

We kindly thank the referee for acknowledging our efforts and pointing out the strengths and weaknesses of the first re-submission. We were pleased that the referee: i) valued our pedagogical approach to unbiased ED-like numerical methods; ii) recognized the advantages in computational performance in our Chebyshev-based approach based on the novel methods introduced in this work; iii) recognized our original results for the signatures of quantum phase transitions in the Kitaev-Ising model obtained from spin dynamics using one of our newly introduced unbiased methods. Below, we address the requested changes, explaining how they were incorporated in our second resubmission, which we believe to overcome the weaknesses of the first resubmission.

1.We achieve remarkable low-temperature stability with our FTCP approach because of the decomposition in a product of exponentials of Equation (34). The inverse temperature steps are kept small enough that the Chebyshev expansions of Equation (36) remain numerically stable, i.e. the argument of the rapidly increasing modified Bessel functions is controlled. In contrast, in the low-temperature regime, the CTPQ expansion of Equation (31) involves rapidly increasing factorial functions of the inverse temperature and the number of iterations. At low temperatures, Equation (31) can no longer be easily evaluated with enough accuracy because the limits of machine precision are reached and eventually it generates overflows. This gradual loss of accuracy becomes visible for the CTPQ results as the temperature is lowered. Eventually, the specific heat diverges, rather than going to zero, as expected. This is inconsistent with the other methods and is merely a numerical artefact due to loss of precision. We added a sentence in Sec. 4.1.2., explaining why FTCP avoids the problem in evaluating the specific heat in contrast to CTPQ: “The operator break-up…”

  1. We agree with the referee. These references now appear in the revised manuscript. The Reference Phys. Rev. B 92, 214431 (2015) is cited in Sec. 3.3.3., in the sentence “We could expand the operator…”. The Reference Phys. Rev. Lett. 121, 220601 (2018) is cited in Sec. 3.4: “On the other hand, the Chebyshev approximation of the time evolution operator --- which is used e.g. in Ref. Phys. Rev. Lett. 121, 220601 (2018) in combination with CTPQ…”. Unlike Phys. Rev. B 92, 214431 (2015), in our finite temperature Chebyshev polynomial (FTCP) approach, we do not Chebyshev-expand the operator exp(-beta H/2) directly. Instead, we first write exp(-beta H/2) as a product over L inverse temperature steps in Eq. (34). Then, we Chebyshev-expand each term of the product. This procedure allows us to maintain numerical stability because the arguments of the fast growing modified Bessel functions of Eq. (35) are kept small. If this is not done, we run into overflows when probing low temperatures. In Phys. Rev. B 92, 214431 (2015), the temperature is kept high enough that this never becomes a problem. For the systems we study in our work, low-temperature features are of particular interest, which motivated us to develop our FTCP method. We added a sentence in Sec. 3.3.3. explaining these differences with respect to the method used in Phys. Rev. B 92, 214431 (2015): “We could expand the operator exp(-beta h / 2) in Chebyshev polynomials directly as done e.g. in Phys. Rev. B 92, 214431 (2015). However, such an expansion is vulnerable to numerical instabilities for large beta (low temperature) due to the rapid growth of the Bessel functions.” In Phys. Rev. Lett. 121, 220601 (2018), the authors use Chebyshev expansions to approximate the time evolution operator. The authors remark that in order to ensure convergence of the spectral functions in their approach, it is necessary to take a long enough time window. Additionally, it is important to consider the time step and the number of polynomials. In contrast, our approach does not rely on time-domain correlators. Instead, we use a Chebyshev expansion to compute the frequency-domain spectral functions directly. The main advantage of our approach is that convergence is set via the number of polynomials. There is no need to worry about a time window, or a time step. Moreover, the authors of Phys. Rev. Lett. 121, 220601 (2018) mention using up to 8000 time steps, with Chebyshev expansions up to order 500. This suggests that our method might be more efficient, possibly due to treating spectral functions directly in the frequency domain. Our approach involves only a single Chebyshev expansion with 3000 polynomials to resolve the spectral function, which is roughly equivalent to 6 time steps using 500 polynomials in Phys. Rev. Lett. 121, 220601 (2018).

  2. We chose the ground state to bench-mark CPGF simply to check consistency with Lanczos ED results. We agree with the referee that Lanczos is the method of choice to study the ground state. Thus, we have expanded our comparison between MCLM and CPGF beyond the ground state. We now target excited states (as explained in Figure 3 in the revised version of the manuscript). In particular, we compare the performance of MCLM and CPGF in a computation of the nearest neighbor spin-spin correlation, similarly to what had been done for the ground state in the previous submission. MCLM and CPGF results are consistent. In the bottom panel of Figure 4 in the revised version, we show the CPGF results, which match MCLM results. Finally, we added a panel to Figure 14, showing our comparison between the computer time required to target the excited states with the two methods. Although CPGF was already faster for the ground state, the difference in performance is more dramatic for excited states (CPGF is about an order of magnitude faster than MCLM). We conclude that CPGF is the method of choice for targeting excited states because of its control over resolution and its advantageous convergence properties.

---

## Round 3 · List of Changes

-The references suggested by the first referee were added in page 3 after the sentence: “The severity of the problem depends on the computational basis…”
-A sentence was added in Sec. 3.3.3. to clarify an aspect of the FTCP method as requested by Referee 2: “We could expand the operator exp(-beta h / 2) in Chebyshev polynomials directly as done e.g. in Phys. Rev. B 92, 214431 (2015). However, such an expansion is vulnerable to numerical instabilities for large beta due to the rapid growth of the Bessel functions.”
-A sentence was slightly modified in Sec. 3.4.: “On the other hand, the Chebyshev approximation of the time evolution operator --- which is used e.g. in Phys. Rev. Lett. 121, 220601 (2018) in combination with CTPQ --- relies on…”. We also added the following footnote “In Phys. Rev. Lett. 121, 220601 (2018), CTPQ is used to generate an initial thermal state at t=0. Time evolution is carried out using the Chebyshev approximation of exp(-i t H)”.
-Figure 3 has been changed slightly so as to show the target energy used to compare MCLM and CPGF. The caption has also been changed to reflect this.
-The bottom panel of Figure 4 now shows CPGF results targeting an excited state. MCLM results match CPGF and thus they are not shown so as not to overcrowd the figure. The caption and legend have also been updated to explain the extra data. A sentence has been added to comment on the excited state data: “Also shown in the bottom panel of Fig. 4…”
-A sentence has been added in Sec. 4.1.1. after Figure 6, commenting on the advantages of using CPGF to target the excited states: “Nevertheless, when targeting the ground state, CPGF is 25% faster on average. For the excited states…”
-A sentence was added in Sec. 4.1.2., reinforcing the numerical stability of FTCP in contrast to CTQP: “The operator break-up…”
-The color bar in Figure 12 has been adapted. The white space indeed means that the spectral function vanishes, as the referee pointed out. This is now evident in the bottom of the color bar, where the blue color is smoothly changed to white near 0. The caption has also been extended to include an explanation of the color bar.
-A sentence was added to the second to last paragraph of Sec. 5, explaining that larger systems can be tackled with ED-like methods by taking advantage of the symmetries of the problem at hand.
-A new panel has been added to Figure 14 with a comparison of the computer time required to target excited states with MCLM and CPGF. The caption has been changed accordingly and the final paragraph of the appendix has also been expanded. We find that CPGF significantly outperforms MCLM. In fact, with CPGF, the overall computational cost of targeting the excited states is comparable to that of targeting the ground state. With MCLM, the cost is larger by about an order of magnitude in comparison to CPGF.

---

## Editorial Decision

published